# Deep Ensemble as a Gaussian Process Approximate Posterior

## Abstract

Deep Ensemble is a flexible and effective alternative to Bayesian neural networks for uncertainty estimation in deep learning. However, Deep Ensemble is broadly criticized for lacking a proper Bayesian justification. Some attempts try to fix this issue, while they are typically coupled with the regression likelihood or rely on restrictive assumptions. In this work, we propose to define a Gaussian process (GP) approximate posterior with Deep Ensemble, based on which we perform variational inference directly in function space. We further develop a function-space posterior regularization mechanism to properly incorporate prior knowledge. We provide strategies to make the training feasible, and demonstrate the algorithmic benefits of variational inference in the GP family. As a result, our method consumes only marginally added training cost than the standard Deep Ensemble. Empirically, our approach achieves better uncertainty estimation than Deep Ensemble and its variants across diverse scenarios.

## 1 Introduction

Bayesian treatment of deep neural networks (DNNs) is promised to enjoy principled Bayesian uncertainty while unleashing the capacity of DNNs, with Bayesian neural networks (BNNs) as popular examples (MacKay, 1992; Hinton & Van Camp, 1993; Neal, 1995; Graves, 2011). Nevertheless, despite the surge of advance in BNNs (Louizos & Welling, 2016; Zhang et al., 2018), many of existing BNNs still face challenges in accurate and scalable inference (Sun et al., 2019), and exhibit limitations in uncertainty estimation and out-of-distribution robustness (Ovadia et al., 2019).

Alternatively, Deep Ensemble (DE) (Lakshminarayanan et al., 2017) trains multiple independent, randomly-initialized DNNs for ensemble, presenting higher flexibility and effectiveness than BNNs. However, it is hard to interpret DE as a Bayesian approach which seeks for the Bayesian posterior of a certain model, and there is no guarantee that the uncertainty estimates given by DE are reliable. To chase a Bayesian justification for DE, recent works like RMS (Lu & Van Roy, 2017; Osband et al., 2018; Pearce et al., 2020) and NTKGP (He et al., 2020) refine DE to be a "sample-then-optimize" approach (Matthews et al., 2017). Nonetheless, these works rely on the *regression likelihood* and often make strong assumptions like *linearised* and/or *infinite-width* models.

To build a Bayesian refinement of DE without reliance on restrictive assumptions, we propose to use DE to define a *Gaussian process* (GP) approximate posterior, which is dubbed as DE-GP for short. The imposition of a GP form on the approximate posterior is empowered by the *equivalence* between GPs and BNNs (Khan et al., 2019; Neal, 1996; Lee et al., 2018; Garriga-Alonso et al., 2018; Matthews et al., 2018; Novak et al., 2018; He et al., 2020), as well as the associated computational benefits when adopting a GP prior without loss of generality.

Concretely, we evaluate the empirical mean and covariance of the ensemble members and use them to specify DE-GP. This is inspired by the formula of neural network Gaussian process (NN-GP) (Novak et al., 2018). Nevertheless, we clarify that we define a parameteric and learnable GP kernel with finitely many and adaptive basis functions, in contrast to the NN-GP kernels built with infinitely many yet fixed basis functions. The flexibility of DE-GP is assured by the expressiveness of DNN ensemble members – by tuning them it is enabled to evolve over a rich variety of kernels.

Given these setups, we perform function-space variational inference (VI) for learning. We maximize the functional evidence lower bound (fELBO) (Sun et al., 2019; Rudner et al., 2021) to push DE-GP towards the true posterior over functions. We further present a posterior regularization scheme in function space to conveniently incorporate structural priors, and present a concrete example on how

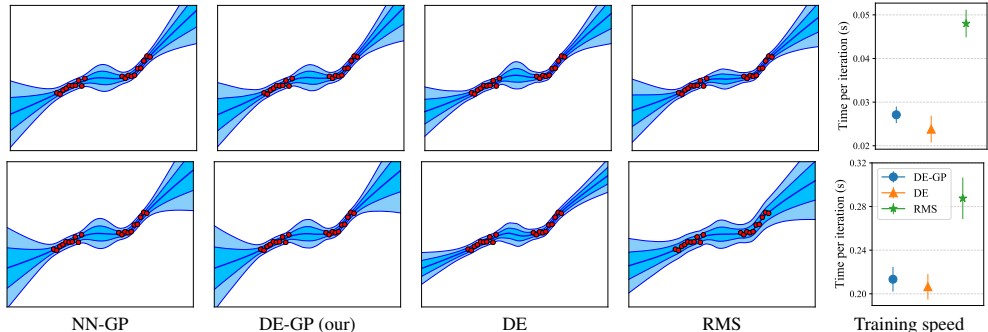

Figure 1: Prediction on data from $y = x^3/4 + \epsilon, \epsilon \sim \mathcal{N}(0, 0.1)$. The two rows correspond to multilayer perceptrons (MLPs) with 2 hidden layers of 64 units and MLPs with 3 hidden layers of 512 units, respectively. The first four columns correspond to four methods. We plot the training data (red dots), mean predictions (dark blue curves), and uncertainty (shaded regions). For NN-GP, we performs analytic GP regression without training MLPs. The other methods train 50 MLPs, with the training speed depicted in the last column. As network size rises, DE-GP behaves consistently, while DE and RMS suffer from degeneracy. DE-GP is on par with the non-parameteric NN-GP in aspect of uncertainty quality, with only marginally added overheads upon DE.

to control function complexity. By VI, we are enabled to handle classification problems directly and exactly, without casting them into regression ones (He et al., 2020). In practice, DE-GP can be easily implemented on top of a standard DE, while only introducing minimally added cost.

The function-space Bayesian inference enables DE-GP to ameliorate the pathologies induced by the over-parameterization nature of DNNs (Sun et al., 2019; Wang et al., 2019). Thus DE-GP can offer calibrated uncertainty estimates, especially when adopting *large* networks (see Fig. 1). We empirically demonstrate that DE-GP outperforms strong baselines on various regression datasets, and presents superior uncertainty estimates and out-of-distribution robustness without compromising accuracy in challenging classification tasks. DE-GP also shows promise in solving contextual bandit problems, where the uncertainty plays a vital role in guiding exploration.

## 2 RELATED WORK

**Bayesian neural networks (BNNs).** Bayesian treatment of DNNs is an emerging topic in deep learning yet with a long history (Mackay, 1992; Hinton & Van Camp, 1993; Neal, 1995; Graves, 2011). BNNs can be learned by virtue of variational inference (Blundell et al., 2015; Hernández-Lobato & Adams, 2015; Louizos & Welling, 2016; Zhang et al., 2018; Khan et al., 2018; Deng et al., 2020), Laplace approximation (Mackay, 1992; Ritter et al., 2018), Markov chain Monte Carlo (Welling & Teh, 2011; Chen et al., 2014; Zhang et al., 2019), particle-optimization based variational inference (Liu & Wang, 2016), Monte Carlo dropout (Gal & Ghahramani, 2016), and other methods (Maddox et al., 2019; Izmailov et al., 2020). To avoid the difficulties of posterior inference in weight space, some recent works advocate performing Bayesian reasoning in function space (Sun et al., 2019; Rudner et al., 2021; Wang et al., 2019). In function space, BNNs of infinite or even finitely width equal to Gaussian processes (GPs) (Neal, 1996; Lee et al., 2018; Novak et al., 2018; He et al., 2020; Khan et al., 2019), which provides valuable insights for this work.

**Deep Ensemble (DE)** (Lakshminarayanan et al., 2017) is a qualified alternative to BNNs for uncertainty estimation (Ovadia et al., 2019), yet lacks a proper Bayesian interpretation. Wilson & Izmailov (2020) conceptually interpreted DE as a method that approximates the Bayesian posterior predictive distribution, but it is hard to judge whether the approximation is reliable or not in practice. Recently, a suite of works based on the notion of "sample-then-optimize" (Matthews et al., 2017) have been developed to make DE Bayesian. For example, RMS (Lu & Van Roy, 2017; Osband et al., 2018; Pearce et al., 2020) regularizes the ensemble members towards randomised priors to obtain posterior samples after training, while it typically assumes *linear* data likelihood which is impractical for deep models and classification tasks. He et al. (2020) proposed to add a randomised function to each ensemble member to realise a function-space Bayesian interpretation, but the method is asymptotically exact in the *infinite width* limit and is limited to regressions. By contrast, DE-GP works without restrictive assumptions. In parallel, D'Angelo & Fortuin (2021) proposed to add a repulsive term to DE to achieve Bayesian inference, which yet relies on unscalable gradient estimators.

## 3 BACKGROUND

### 3.1 BAYESIAN NEURAL NETWORKS

Consider learning on a dataset $\mathcal{D} = (\mathbf{X}, \mathbf{Y}) = \{(\boldsymbol{x}_i, \boldsymbol{y}_i)\}_{i=1}^n$, with $\boldsymbol{x}_i \in \mathcal{X}$ as data and $\boldsymbol{y}_i$ as $C$-dimensional targets. Let $g(\cdot, \boldsymbol{w}) : \mathcal{X} \to \mathbb{R}^C$ denote the function represented by a DNN with weights $\boldsymbol{w}$. Bayesian neural networks (BNNs) treat $\boldsymbol{w}$ as random variables and chase the posterior $p(\boldsymbol{w}|\mathcal{D})$ by imposing a prior $p(\boldsymbol{w})$. Among the family, variational BNNs (VBNNs) are particularly appealing as they cast posterior inference as an optimization problem (Graves, 2011; Blundell et al., 2015; Louizos & Welling, 2016), where a variational distribution $q(\boldsymbol{w})$ is introduced to approximate the true posterior by maximizing the evidence lower bound (ELBO):

$$\mathbb{E}_{q(\boldsymbol{w})}[\log p(\mathcal{D}|\boldsymbol{w})] - D_{\mathrm{KL}}[q(\boldsymbol{w})\|p(\boldsymbol{w})]. \tag{1}$$

BNNs marginalize over the posteriors to obtain the *posterior predictive* for new data $\boldsymbol{x}^*$:

$$p(y|\boldsymbol{x}^*, \mathcal{D}) = \mathbb{E}_{p(\boldsymbol{w}|\mathcal{D})} p(y|\boldsymbol{x}^*, \boldsymbol{w}) \approx \mathbb{E}_{q(\boldsymbol{w})} p(y|\boldsymbol{x}^*, \boldsymbol{w}) \approx \sum_{s=1}^S p(y|\boldsymbol{x}^*, \boldsymbol{w}_s), \tag{2}$$

where $\boldsymbol{w}_s \sim q(\boldsymbol{w}), s = 1, ..., S$. Such a procedure is expected to propagate the embedded model uncertainty into the prediction. However in practice, most of the existing BNN approaches face obstacles in precise posterior inference due to non-trivial and convoluted posterior dependencies (Louizos & Welling, 2016; Zhang et al., 2018; Shi et al., 2018; Sun et al., 2019), and deliver unsatisfactory uncertainty estimation and out-of-distribution (OOD) robustness (Ovadia et al., 2019).

### 3.2 DEEP ENSEMBLE

As a workaround of BNNs, Deep Ensemble (DE) (Lakshminarayanan et al., 2017) deploys a set of $M$ DNNs $\{g(\cdot, \boldsymbol{w}_i)\}_{i=1}^M$ for ensemble. $\{\boldsymbol{w}_i\}_{i=1}^M$ are independently trained to interpret the data from different angles, under maximum likelihood estimation (MLE) principle in standard DE, or maximum a posteriori (MAP) principle in regularized DE (rDE):

$$\max_{\boldsymbol{w}_1, ..., \boldsymbol{w}_M} \mathcal{L}_{\mathrm{DE}} = \frac{1}{M} \sum_{i=1}^M \log p(\mathcal{D}|\boldsymbol{w}_i), \quad \max_{\boldsymbol{w}_1, ..., \boldsymbol{w}_M} \mathcal{L}_{\mathrm{rDE}} = \frac{1}{M} \sum_{i=1}^M [\log p(\mathcal{D}|\boldsymbol{w}_i) + \log p(\boldsymbol{w}_i)]. \tag{3}$$

Due to the randomness in network initialization and stochastic gradient descent (SGD), DE can effectively explore the non-convex, multimodal loss landscape of DNNs (Fort et al., 2019; Wilson & Izmailov, 2020), thus has shown promise in various uncertainty estimation tasks (Lakshminarayanan et al., 2017; Ovadia et al., 2019). However, due to the lack of a Bayesian justification, there is no guarantee that the uncertainty estimates given by DE are reliable.

### 3.3 RELATION OF THE BAYESIAN POSTERIORS OF DNNs TO GAUSSIAN PROCESSES

It is early shown that shallow BNNs converge to Gaussian processes (GPs) in the infinite width limit (Neal, 1996). Recently, there is ongoing effort to extend the result to multiple layers (Lee et al., 2018; Matthews et al., 2018), convolutional architectures (Garriga-Alonso et al., 2018; Novak et al., 2018), and beyond (Yang, 2019). Briefly, an infinitely wide BNN amount to the NN-GP:

$$\mathcal{GP}(0, \kappa(\boldsymbol{x}, \boldsymbol{x}')), \text{ with } \kappa(\boldsymbol{x}, \boldsymbol{x}') := \mathbb{E}_{\boldsymbol{w} \sim p(\boldsymbol{w})}[g(\boldsymbol{x}, \boldsymbol{w}) g(\boldsymbol{x}', \boldsymbol{w})^\top], \tag{4}$$

where $p(\boldsymbol{w})$ is a layerwise isotropic Gaussian and the DNN specifying $g$ has infinite width. Notably, $\kappa$ is a compositional, matrix-valued kernel, with values in the space of $C \times C$ matrices, and can be analytically estimated in some cases (Novak et al., 2018; 2019). Due to the i.i.d. weights, the kernel evaluation $\kappa(\boldsymbol{x}^*, \boldsymbol{x}^*)$ of some data $\boldsymbol{x}^*$ is a scaled identity matrix.

Based on the canonical results of GP regression, we know the posteriors of infinitely wide BNNs are still GPs (in regression tasks), corresponding to the Bayesian training of the *last* readout layer (Arora et al., 2019; Lee et al., 2019). Despite a golden standard for small regression problems, analytical posterior inference for NN-GP faces challenges when handling contemporary DNN architectures (Novak et al., 2018), large data (Shi et al., 2019) and classification problems.

Interestingly, finitely wide BNNs with Gaussian posteriors on weights can also be connected to GPs (Khan et al., 2019; Rudner et al., 2021). We then arrive at the hypothesis that the Bayesian posteriors of DNNs are closely related to GPs. So, we specify a GP approximate posterior with DE.

## 4 METHODOLOGY

### 4.1 DEEP ENSEMBLE AS A GAUSSIAN PROCESS APPROXIMATE POSTERIOR

Basically, we utilize DE's ensemble members (referred to as *basis functions* hereinafter) to define a GP approximate posterior as follows:

$$q(f; \boldsymbol{w}_1, ..., \boldsymbol{w}_M) := \mathcal{GP}(f|m_q(\boldsymbol{x}), k_q(\boldsymbol{x}, \boldsymbol{x}')),$$

$$m_q(\boldsymbol{x}) := \frac{1}{M} \sum_{i=1}^{M} g_i(\boldsymbol{x}), \tag{5}$$

$$k_q(\boldsymbol{x}, \boldsymbol{x}') := \frac{1}{M} \sum_{i=1}^{M} (g_i(\boldsymbol{x}) - m_q(\boldsymbol{x})) (g_i(\boldsymbol{x}') - m_q(\boldsymbol{x}'))^\top + \lambda \mathbf{I}_C,$$

where $g_i$ refers to $g(\cdot, \boldsymbol{w}_i)$. Namely, the empirical mean and central covariance of finitely many, adaptive basis functions are leveraged to specify DE-GP. $k_q(\boldsymbol{x}, \boldsymbol{x}')$ is a *linear*, *matrix-valued* kernel with a small scaled identity matrix $\lambda \mathbf{I}_C$ appended to avoid singularity. The variations in $k_q(\boldsymbol{x}, \boldsymbol{x}')$ are confined to having up to $M - 1$ rank, echoing the recent investigations showing that low-rank approximate posteriors for DNNs conjoin effectiveness and *efficiency* (Maddox et al., 2019; Izmailov et al., 2020; Dusenberry et al., 2020). Note that the popular Nyström method (Williams & Seeger, 2001) also uses a low-rank matrix to approximate the original kernel matrix.

Akin to the kernels in (Wilson et al., 2016), the DE-GP kernels are highly flexible, and may automatically discover the underlying structures of high-dimensional data without manual participation.

### 4.2 VARIATIONAL INFERENCE IN FUNCTION SPACE

We do function-space variational inference to push DE-GP towards the true posterior over functions.

**Prior** We have high freedom to determine which prior to use attributed to the variational inference paradigm. Without loss of generality, we use the MC estimate of NN-GP (MC NN-GP) (Novak et al., 2018) as the prior due to its accessibility. Concretely, supposing a *finitely wide* DNN composed of a feature projector $h(\cdot, \boldsymbol{w}) : \mathcal{X} \rightarrow \mathbb{R}^{\hat{C}}$ and a linear readout layer with weight variance $\sigma_w^2$ and bias variance $\sigma_b^2$, the MC NN-GP is defined as $p(f) = \mathcal{GP}(f|0, k(\boldsymbol{x}, \boldsymbol{x}'))$, where

$$k(\boldsymbol{x}, \boldsymbol{x}') = (\sigma_w^2 \hat{k}(\boldsymbol{x}, \boldsymbol{x}') + \sigma_b^2) \mathbf{I}_C, \text{with } \hat{k}(\boldsymbol{x}, \boldsymbol{x}') = \frac{1}{S\hat{C}} \sum_{s=1}^{S} h(\boldsymbol{x}, \boldsymbol{w}_s)^\top h(\boldsymbol{x}', \boldsymbol{w}_s). \tag{6}$$

$\mathbf{I}_C$ refers to the indentity matrix of size $C \times C$ and $\boldsymbol{w}_s$ are i.i.d. samples from the Gaussian prior on weights $p(\boldsymbol{w})$. Similar setups can be found in some related works like (Wang et al., 2019).

There may be a subtle difference between the MC NN-GP and the NN-GP, so what we are actually doing is not approximating the exact NN-GP posterior with DE-GP. Also of note that the NN-GP prior and the DE-GP posterior can be defined with various architectures.

**fELBO** Following Sun et al. (2019), we maximize the functional ELBO (fELBO) to optimize DE-GP:

$$\max_{q(f)} \mathbb{E}_{q(f)}[\log p(\mathcal{D}|f)] - D_{\text{KL}}[q(f)\|p(f)]. \tag{7}$$

Notably, there is a KL divergence between two GPs, which, on its own, is challenging to cope with. Fortunately, as proved by Sun et al. (2019), we can take the KL divergence between the marginal distributions of function evaluations as a substitute for it, giving rise to a more tractable objective:

$$\max_{q(f)} \mathcal{L} = \sum_{(\boldsymbol{x}_i, \boldsymbol{y}_i) \in \mathcal{D}} \mathbb{E}_{q(f)}[\log p(\boldsymbol{y}_i|f(\boldsymbol{x}_i))] - D_{\text{KL}}[q(\mathbf{f}^{\tilde{\mathbf{X}}})\|p(\mathbf{f}^{\tilde{\mathbf{X}}})], \tag{8}$$

where $\tilde{\mathbf{X}}$ denotes a measurement set including all training inputs $\mathbf{X}$, and $\mathbf{f}^{\tilde{\mathbf{X}}}$ is the concatenation of the vectorized outputs of $f$ for $\tilde{\mathbf{X}}$, i.e., $\mathbf{f}^{\tilde{\mathbf{X}}} \in \mathbb{R}^{|\tilde{\mathbf{X}}|C}$.

### 4.3 POSTERIOR REGULARIZATION IN FUNCTION SPACE

Though Bayesian learning in function space enables a direct imposition of prior knowledge on function properties like smoothness and periodicity (Sun et al., 2019), sometimes the prior knowledge cannot be trivially designed as a prior distribution over functions. For example, the functions are

periodical only over a limited input range, etc. In these cases, posterior regularization (Ganchev et al., 2010; Zhu et al., 2014) provides a principled workaround to impose structural constraints.

Briefly, we apply posterior regularization to functional variational inference by solving:

$$\max_{q(f)} \mathcal{L} \text{ s.t.: } q(f) \in Q. \tag{9}$$

$Q = \{q(f)|\mathbb{E}_{q(f)}\Omega(f) \leq 0\}$ is a valid set defined in terms of a functional $\Omega$ which delivers some statistic of interest of a function.[1] For tractable optimization, we slack the constraint as a penalty:

$$\max_{q(f)} \mathcal{L}' = \mathcal{L} - \beta \max\{\mathbb{E}_{q(f)}\Omega(f), 0\}, \tag{10}$$

where $\beta$ is a trade-off coefficient.

We next present an example on how to impose prior knowledge on function (hypothesis) complexity on the DE-GP approximate posterior using this paradigm.[2] For the sake of brevity, we assume a binary classification scenario in the following discussion where $y \in \{-1, 1\}$ and $f, g_i : \mathcal{X} \to \mathbb{R}$. We use 0-1 loss $\ell(f(\boldsymbol{x}), y) = \mathbf{1}_{y \neq \text{sign}(f(\boldsymbol{x}))}$ to measure the classification error on one datum. We assume an underlying distribution $\mu = \mu(\boldsymbol{x}, y)$ supported on $\mathcal{X} \times \{-1, 1\}$ for generating the training data $\mathcal{D}$, based on which we can define the *true* risk of a function (hypothesis) $f$: $R(f) := \mathbb{E}_{(\boldsymbol{x},y)\sim\mu}\ell(f(\boldsymbol{x}), y)$. We set $\mathbb{E}_{q(f)}\Omega(f) := \mathbb{E}_{q(f)}R(f)$ in the seek of a posterior over functions that can generalize well.

By definition, a hypothesis sample $f \sim q(f) = \mathcal{GP}(m_q(\boldsymbol{x}), k_q(\boldsymbol{x}, \boldsymbol{x}'))$ can be decomposed as $f(\boldsymbol{x}) = \frac{1}{M}\sum_{i=1}^{M} g_i(\boldsymbol{x}) + \zeta(\boldsymbol{x})$ with $\zeta(\boldsymbol{x}) \sim \mathcal{GP}(0, k_q(\boldsymbol{x}, \boldsymbol{x}'))$. If $\text{sign}(f(\boldsymbol{x})) \neq y$, it is impossible that $\text{sign}(g_1(\boldsymbol{x})) = y, ..., \text{sign}(g_M(\boldsymbol{x})) = y$, and $\text{sign}(\zeta(\boldsymbol{x})) = y$ all hold. In other words,

$$\ell(f(\boldsymbol{x}), y) \leq \sum_{i=1}^{M}[\ell(g_i(\boldsymbol{x}), y)] + \ell(\zeta(\boldsymbol{x}), y). \tag{11}$$

We can further re-parameterize $\zeta(\boldsymbol{x})$ as $\zeta(\boldsymbol{x}) = \frac{1}{\sqrt{M}}\sum_{i=1}^{M} \epsilon_i(g_i(\boldsymbol{x}) - m_q(\boldsymbol{x})) + \sqrt{\lambda}\epsilon_0, \epsilon_i \sim \mathcal{N}(0, 1), i = 0, ..., M$, which is essentially a real-valued random function symmetric around 0. Thus, for any $(\boldsymbol{x}, y) \sim \mu$, we have $\mathbb{E}_{q(f)}\ell(\zeta(\boldsymbol{x}), y) = \mathbb{E}_{\epsilon_0,...,\epsilon_M}\ell(\zeta(\boldsymbol{x}), y) = 1/2$. As a result,

$$\mathbb{E}_{q(f)}R(f) \leq \mathbb{E}_{q(f)}\mathbb{E}_{(\boldsymbol{x},y)\sim\mu} \sum_{i=1}^{M}[\ell(g_i(\boldsymbol{x}), y)] + \mathbb{E}_{(\boldsymbol{x},y)\sim\mu}[1/2] = \sum_{i=1}^{M}[R(g_i)] + 1/2. \tag{12}$$

Namely, the expected generalization error of the approximately posteriori functions can be bounded from above by those of the DNN basis functions.[3] Recalling the theoretical and empirical results showing that DNNs' generalization error $R(g_i)$ can be decreased by controlling model capacity in terms of norm-based regularization $\min_{\boldsymbol{w}_i} ||\boldsymbol{w}_i||_2^2$ (Neyshabur et al., 2015; 2017; Bartlett et al., 2017; Jiang et al., 2019), we opt to maximize the following refined fELBO for learning DE-GP:

$$\max_{\boldsymbol{w}_1,...,\boldsymbol{w}_M} \mathcal{L}_{\text{DE-GP}} = \sum_{(\boldsymbol{x}_i,\boldsymbol{y}_i)\in\mathcal{D}} \mathbb{E}_{q(f)}[\log p(\boldsymbol{y}_i|f(\boldsymbol{x}_i))] - D_{\text{KL}}[q(\mathbf{f}^{\tilde{\mathbf{X}}})||p(\mathbf{f}^{\tilde{\mathbf{X}}})] - \beta\sum_i ||\boldsymbol{w}_i||_2^2. \tag{13}$$

Intuitively, the weight space of an over-parameterized DNN contains many redundant interpretations for the data, thus pushing the weights towards zero would not hurt data fitting but boost generalization. In practice, when processing with shallow architectures, $\beta$ can be set as 0 to remove the constraint on model capacity; while when using deep architectures (e.g., ResNet (He et al., 2016)), we can set $\beta$ according to commonly used weight decay coefficient for better generalization. Comparisons in Fig. 2 and Table 1 support this.

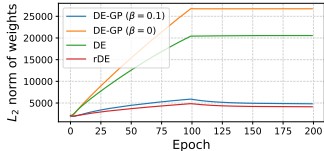

Figure 2: Comparison on the $L_2$ norm of weights. We traine models on CIFAR-10 with ResNet-20. DE-GP ($\beta = 0$) finds solutions with high complexity and poor test accuracy (see Table 1), yet DE-GP ($\beta = 0.1$) settles this.

### 4.4 THE ALGORITHM

We outline the training procedure of DE-GP in Algorithm 1, and elaborate some details below.

---

[1] Here we assume one-dimensional outputs for $\Omega$ for notation compactness.

[2] The motivation is that, as observed, the fELBO cannot cause proper regularization effects on weights $\boldsymbol{w}_i$ for deep architectures, thus the learned DE-GP may suffer from high-complexity basis functions.

[3] A similar conclusion can be drawn in multi-class classification scenario (see Appendix A).

---

**Algorithm 1:** Deep Ensemble as a Gaussian process posterior (DE-GP).

1: **Input:** $\mathcal{D}$: dataset; $\{g_i\}_{i=1}^M$: $M$ DNNs; $k$: prior kernel; $\alpha, \beta$: trade-off coefficients; $\nu$: distribution for sampling extra measurement points; $U$: number of MC samples for estimating the expected log-likelihood
2: **while** not converged **do**
3:    $\mathcal{D}_s = (\mathbf{X}_s, \mathbf{Y}_s) \subset \mathcal{D}, \mathbf{X}_\nu \sim \nu, \tilde{\mathbf{X}}_s = \{\mathbf{X}_s, \mathbf{X}_\nu\}$
4:    $\mathbf{g}_i = g_i(\tilde{\mathbf{X}}_s) \in \mathbb{R}^{|\tilde{\mathbf{X}}_s|C}, i = 1, ..., M$
5:    $\mathbf{m}_q^{\tilde{\mathbf{X}}_s} = \frac{1}{M}\sum_i \mathbf{g}_i, \mathbf{k}_q^{\tilde{\mathbf{X}}_s, \tilde{\mathbf{X}}_s} = \frac{1}{M}\sum_{i=1}^M (\mathbf{g}_i - \mathbf{m}_q^{\tilde{\mathbf{X}}_s})(\mathbf{g}_i - \mathbf{m}_q^{\tilde{\mathbf{X}}_s})^\top + \lambda \mathbf{I}_{|\tilde{\mathbf{X}}_s|C}$
6:    $\mathbf{k}^{\tilde{\mathbf{X}}_s, \tilde{\mathbf{X}}_s} = k(\tilde{\mathbf{X}}_s, \tilde{\mathbf{X}}_s) \in \mathbb{R}^{|\mathbf{X}_s|C \times |\mathbf{X}_s|C}$
7:    $\mathcal{L}_1 = \frac{1}{U}\sum_{i=1}^U \sum_{(\boldsymbol{x}, \boldsymbol{y}) \in \mathcal{D}_s} \log p(\boldsymbol{y}|\mathbf{f}_i(\boldsymbol{x})), \mathbf{f}_i \sim \mathcal{N}(\mathbf{m}_q^{\tilde{\mathbf{X}}_s}, \mathbf{k}_q^{\tilde{\mathbf{X}}_s, \tilde{\mathbf{X}}_s})$
8:    $\mathcal{L}_2 = D_{\mathrm{KL}}[\mathcal{N}(\mathbf{m}_q^{\tilde{\mathbf{X}}_s}, \mathbf{k}_q^{\tilde{\mathbf{X}}_s, \tilde{\mathbf{X}}_s}) \| \mathcal{N}(\mathbf{0}, \mathbf{k}^{\tilde{\mathbf{X}}_s, \tilde{\mathbf{X}}_s})], \mathcal{L}_3 = \sum_i \|\boldsymbol{w}_i\|_2^2$
9:    $\boldsymbol{w}_i = \boldsymbol{w}_i + \eta \nabla_{\boldsymbol{w}_i}(\mathcal{L}_1 - \alpha\mathcal{L}_2 - \beta\mathcal{L}_3), i = 1, ..., M$

---

### 4.4.1 MINI-BATCH TRAINING

In deep learning scenarios, DE-GP should proceed by mini-batch training. At each step, we manufacture a stochastic measurement set with a mini-batch $\mathcal{D}_s = (\mathbf{X}_s, \mathbf{Y}_s)$ from the training data $\mathcal{D}$ and random samples $\mathbf{X}_\nu$ from a continuous distribution $\nu$ (e.g., a uniform distribution) supported on $\mathcal{X}$. Then, we adapt the objective defined in Eq. (13) to the following form:

$$\mathcal{L}_{\text{DE-GP}} = \sum_{(\boldsymbol{x}_i, \boldsymbol{y}_i) \in \mathcal{D}_s} \mathbb{E}_{q(f)}[\log p(\boldsymbol{y}_i|f(\boldsymbol{x}_i))] - \alpha D_{\mathrm{KL}}[q(\mathbf{f}^{\tilde{\mathbf{X}}_s}) \| p(\mathbf{f}^{\tilde{\mathbf{X}}_s})] - \beta \sum_i \|\boldsymbol{w}_i\|_2^2, \quad (14)$$

where $\tilde{\mathbf{X}}_s$ indicates the union of $\mathbf{X}_s$ and $\mathbf{X}_\nu$. When $\alpha = 1$ and $\beta = 0$, $\mathcal{L}_{\text{DE-GP}}$ is a lower bound of $\log p(\mathcal{D}_s)$ according to (Sun et al., 2019). Yet, given that we do not perform intensive hyper-parameter tuning for the NN-GP prior, it is reasonable to make $\alpha$ a tunable hyper-parameter to better trade off between the data fitting and the priori inductive bias.

### 4.4.2 AN EXACT AND EFFICIENT ESTIMATION OF THE MARGINAL KL DIVERGENCE

The marginal distributions of function evaluations are multivariate Gaussian by the definition of GP:

$$q(\mathbf{f}^{\tilde{\mathbf{X}}_s}) = \mathcal{N}(\mathbf{f}^{\tilde{\mathbf{X}}_s}|\mathbf{m}_q^{\tilde{\mathbf{X}}_s}, \mathbf{k}_q^{\tilde{\mathbf{X}}_s, \tilde{\mathbf{X}}_s}), \; p(\mathbf{f}^{\tilde{\mathbf{X}}_s}) = \mathcal{N}(\mathbf{f}^{\tilde{\mathbf{X}}_s}|\mathbf{0}, \mathbf{k}^{\tilde{\mathbf{X}}_s, \tilde{\mathbf{X}}_s}), \quad (15)$$

with the kernel matrices $\mathbf{k}_q^{\tilde{\mathbf{X}}_s, \tilde{\mathbf{X}}_s}, \mathbf{k}^{\tilde{\mathbf{X}}_s, \tilde{\mathbf{X}}_s} \in \mathbb{R}^{|\tilde{\mathbf{X}}_s|C \times |\tilde{\mathbf{X}}_s|C}$ as the joints of pair-wise outcomes.

Therefore, the marginal KL divergence and its gradients can be estimated exactly without resorting to some approximations (Rudner et al., 2021). What's more, as discussed in Section 3.3, there is a simple structure in the prior kernel matrices, so we can write them in the form of Kronecker product:

$$\mathbf{k}^{\tilde{\mathbf{X}}_s, \tilde{\mathbf{X}}_s} \approx (\sigma_w^2 \hat{\mathbf{k}}^{\tilde{\mathbf{X}}_s, \tilde{\mathbf{X}}_s} + \sigma_b^2) \otimes \mathbf{I}_C, \quad (16)$$

where $\hat{\mathbf{k}}^{\tilde{\mathbf{X}}_s, \tilde{\mathbf{X}}_s} \in \mathbb{R}^{|\tilde{\mathbf{X}}_s| \times |\tilde{\mathbf{X}}_s|}$ corresponds to the evaluation of kernel $\hat{k}$. Hence we can exploit the property of Kronecker product to inverse $\mathbf{k}^{\tilde{\mathbf{X}}_s, \tilde{\mathbf{X}}_s}$ in $\mathcal{O}(|\tilde{\mathbf{X}}_s|^3)$ complexity.

Besides, as $\mathbf{k}_q^{\tilde{\mathbf{X}}_s, \tilde{\mathbf{X}}_s}$ is low-rank, we can leverage the Woodbury matrix identity (Woodbury, 1950) and matrix determinant lemma (Harville, 1998) to efficiently compute the inverse and determinant of $\mathbf{k}_q^{\tilde{\mathbf{X}}_s, \tilde{\mathbf{X}}_s}$ in $\mathcal{O}(|\tilde{\mathbf{X}}_s|CM^2)$ time complexity given that usually $M \ll |\tilde{\mathbf{X}}_s|C$ (e.g., $10 \ll 256C$).

## 4.5 DISCUSSION

**Diversity.** The diversity among the ensemble members in function space is explicitly encouraged by the KL divergence between variational and the prior in fELBO. Nonetheless, the expected log-likelihood in fELBO enforces each ensemble member to yield the same, correct outcomes for the training data. Thereby, the diversity mainly exists in the regions far away from the training data (see Fig. 1 and Fig. 9 in Appendix B.2). Yet, the diversity in DE does not have a clear theoretical support.

**Efficiency.** Compared to the overhead introduced by DNNs, the effort of estimating the KL divergence between Gaussians is negligible. The added cost of DE-GP primarily arises from the extra measurement points and the evaluation of the prior kernels. In practice, we use a small batch size

Figure 3: Comparison on average test NLL and RMSE on UCI regression problems. The lower the better.

for the extra measurement points. We build the prior NN-GP kernels with cheap architectures and perform MC estimation in parallel. Eventually, DE-GP is only marginally slower than DE in training.

**Weight sharing.** DE-GP does not care about how the basis functions are parameterized, so we can perform weight sharing among the basis functions, for example, using a shared feature extractor and $M$ independent MLP classifiers to construct $M$ basis functions (Deng et al., 2021). With shared weights, DE-GP is still likely to be reliable because our learning principle induces diversity in function space regardless of the weights. Experiments in Section 5.3 validate this.

**Limitations.** Despite being Bayesian in principle, DE-GP is likely to be less flexible than DE and other variants because all the involved basis functions need to be updated simultaneously.

## 5 EXPERIMENTS

We perform extensive evaluation to prove that DE-GP yields better uncertainty estimates than the baselines, while preserving non-degraded predictive performance. The baselines include DE, rDE, NN-GP, RMS, etc. In *all* experiments, we estimate the prior kernel with 10 MC samples and set the sampling distribution for extra measurement points $\nu$ as the uniform distribution over the data region.

### 5.1 ILLUSTRATIVE REGRESSION

We build two regression problems with data from $y = x^3/4 + \epsilon, \epsilon \sim \mathcal{N}(0, 0.1)$ and $y = x \sin 5x + \epsilon, \epsilon \sim \mathcal{N}(0, 0.2)$, respectively. We consider two architectures: MLPs with 2 hidden layers of 64 units and MLPs with 3 hidden layers of 512 units, where ReLU activation is used. For NN-GP, we analytically estimate the GP kernel and perform GP regression without training MLPs. For DE-GP, DE, rDE, and RMS, we train 50 MLPs. For MC dropout, we train a MLP with the commonly used 0.2 dropout rate. We use the default values for $\alpha$ and $\beta$, i.e., $\alpha = 1$ and $\beta = 0$.

Fig. 1 and Appendix B.2 show the results on the two problems. We also provide the comparison on training speed in Fig. 1. As shown, DE-GP delivers calibrated uncertainty estimates across settings, on par with the non-parametric Bayesian baseline NN-GP. Yet, the baselines like DE, rDE, and RMS suffer from degeneracy issue as the dimension of weights increases. Though NN-GP outperforms other methods, the involved analytical GP regression may have scalability and effectiveness issues when facing modern architectures (Novak et al., 2018), while DE-GP does not suffer from them.

### 5.2 UCI REGRESSION

We then assess DE-GP on 5 UCI real-valued regression problems. The used architecture is a MLP with 2 hidden layers of 256 units and ReLU activation. 10 networks are trained for DE, DE-GP and other variants. For DE-GP, we set $\beta = 0$ and tune $\alpha$ according to validation sets.

We perform cross validation with 5 splits. Fig. 3 shows the results. DE-GP surpasses or approaches the baselines across scenarios in aspects of both test negative log-likelihood (NLL) and test root mean square error (RMSE). DE-GP even beats NN-GP, which is probably attributed to that the variational family specified by DE enjoys the beneficial inductive bias of practically sized SGD-trained DNNs, and DE-GP can flexibly trade off between the likelihood and the prior by tuning $\alpha$.

### 5.3 IMAGE CLASSIFICATION ON FASHION-MNIST AND CIFAR-10

In the classification experiments, we augment the data log-likelihood (i.e., the first term in Eq. (14)) with a *trainable temperature* to tackle oversmoothing and avoid underconfidence. [4]

**Fashion-MNIST**. We use a widened LeNet5 architecture with batch normalizations (BNs) (Ioffe & Szegedy, 2015) for the Fashion-MNIST dataset (Xiao et al., 2017). Considering the inefficiency of

---

[4]We can make the temperature a Bayesian variable, but it is unnecessary as our model is already Bayesian.

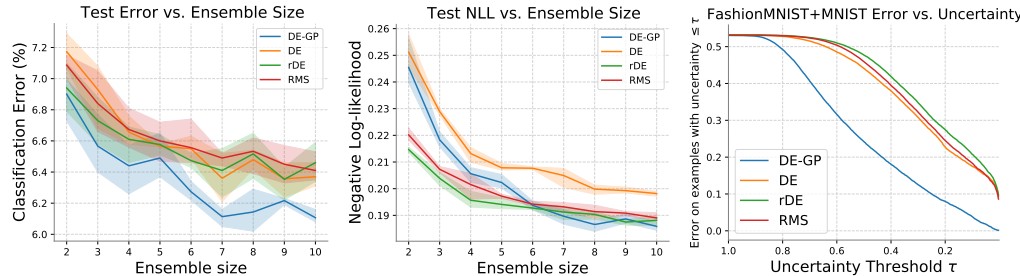

Figure 4: (Left): Test error varies w.r.t. ensemble size on Fashion-MNIST. (Middle): Test NLL varies w.r.t. ensemble size on Fashion-MNIST. (Right): Test error versus uncertainty plots for methods trained on Fashion-MNIST and tested on both Fashion-MNIST and MNIST. Ensemble size is fixed as 10.

Table 1: Test accuracy comparison on CIFAR-10. Results are summarized over 8 trials.

| Architecture | DE-GP ($\beta = 0.1$) | DE-GP ($\beta = 0$) | DE | rDE | RMS |
|---|---|---|---|---|---|
| ResNet-20 | **94.67**±0.04% | 93.71±0.06% | 93.43±0.08% | **94.58**±0.05% | 93.63±0.07% |
| ResNet-56 | **95.55**±0.04% | 94.24±0.07% | 94.04±0.07% | **95.56**±0.06% | 94.45±0.03% |

NN-GP, we mainly compare DE-GP to DE, rDE, and RMS. We set $\alpha$ as well as the regularization coefficients for rDE and RMS all as 0.1 according to validation accuracy. For DE-GP, we use $\beta = 0$ given the limited capacity of the architecture. The in-distribution performance is averaged over 3 runs. Fig. 4-(Left) and Fig. 4-(Middle) display how ensemble size impacts the test results. Surprisingly, the test error of DE-GP is even lower than the baselines.

Besides, to compare the quality of uncertainty estimates, we use the trained models to make prediction and quantify *epistemic uncertainty* for both the in-distribution test set and the out-of-distribution (OOD) MNIST test set. All predictions on OOD data are regarded as wrong. The *epistemic uncertainty* is estimated by the mutual information between the prediction and the variable function:

$$\mathcal{I}(f, y|\boldsymbol{x}, \mathcal{D}) \approx H\left(\frac{1}{S}\sum_{s=1}^{S} p(y|f_s(\boldsymbol{x}))\right) - \frac{1}{S}\sum_{s=1}^{S} H\left(p(y|f_s(\boldsymbol{x}))\right), s = 1, ..., S, \qquad (17)$$

where $H$ indicates Shannon entropy, with $f_s = g(\cdot, \boldsymbol{w}_s)$ for DE, rDE, and RMS, and $f_s \sim q(f; \boldsymbol{w}_1, ..., \boldsymbol{w}_M)$ for DE-GP. We normalize the uncertainty estimates into $[0, 1]$. For each threshold $\tau \in [0, 1]$, we plot the average test error for data with $\leq \tau$ uncertainty in Fig. 4-(Right). It is prominent that under various uncertainty thresholds, DE-GP makes fewer mistakes than the baselines, implying that DE-GP succeeds to assign relatively higher uncertainty for the OOD data.

**CIFAR-10**. Next, we apply DE-GP to the real-world image classification task CIFAR-10 (Krizhevsky et al., 2009). We consider the popular ResNet architectures (He et al., 2016) including ResNet-20 and ResNet-56. The ensemble size is fixed as 10. We split the data as training set, validation set, and test set of size 45000, 5000, 10000, respectively. We set $\beta = 0.1$, equivalent with the regularization coefficient on weight in rDE. We set $\alpha = 0.1$ according to an ablation study in Appendix B.5. We use a lite ResNet-20 architecture without BNs and residual connections to set up the prior NN-GP kernel for both the ResNet-20 and ResNet-56 based variational posteriors.

We present the in-distribution test accuracy in Table 1 and the error versus uncertainty plots on the combination of CIFAR-10 and SVHN test sets in Fig. 5. It is noteworthy that DE-GP is on par with the practically-used, competing rDE in aspect of test accuracy. DE-GP ($\beta = 0$) shows unsatisfactory test accuracy, verifying the necessity of performing posterior regularization to penalize function complexity when using ultra-deep networks. The error versus uncertainty plots are similar to those for Fashion-MNIST, substantiating the universality of DE-GP.

We further test the trained methods on CIFAR-10 corruptions (Hendrycks & Dietterich, 2018), a challenging OOD generalization/robustness benchmark for deep models. As shown in Fig. 6 and Appendix B.3, DE-GP reveals smaller Expected Calibration Error (ECE) (Guo et al., 2017) and lower NLL at various levels of skew, reflecting its ability to make conservative predictions under corruptions and good OOD robustness.

More results for the deeper ResNet-110 architecture and the more challenging CIFAR-100 benchmark are provided in Appendix B.3 and Appendix B.4.

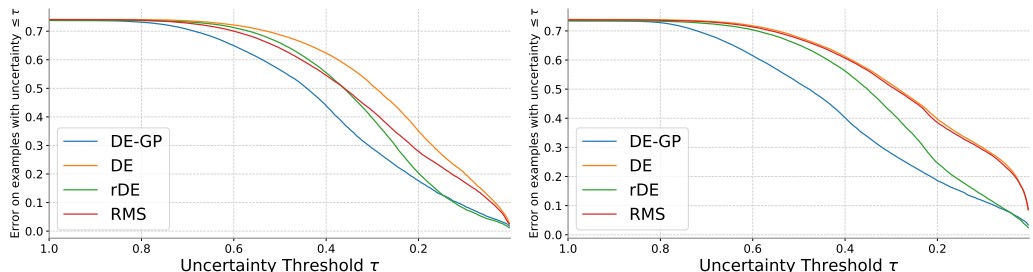

Figure 5: Test error versus uncertainty plots for methods trained on CIFAR-10 and tested on both CIFAR-10 and SVHN with ResNet-20 (Left) or ResNet-56 (Right) architecture.

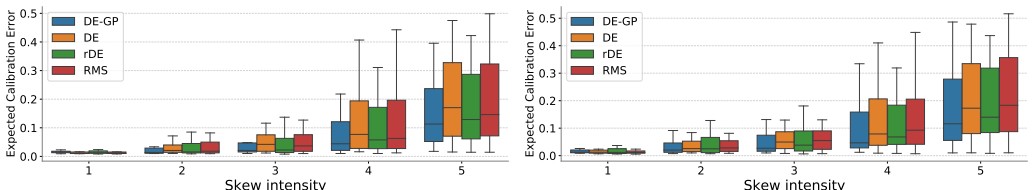

Figure 6: Expected Calibration Error on CIFAR-10 corruptions for models trained with ResNet-20 (Left) or ResNet-56 (Right) architecture. We summarize the results across 19 types of skew in each box.

**Weight Sharing.** We build a ResNet-20 with 10 classification heads and a shared feature extraction module to evaluate the methods under weight sharing. We set a larger value for $\alpha$ for DE-GP to induce higher magnitudes of functional diversity. The test accuracy (over 6 trials) and error versus uncertainty plots on CIFAR-10 are illustrated in Fig. 7. We exclude RMS from the comparison as it assumes i.i.d. basis functions which may be incompatible with weight sharing. DE-GP benefits from the diversity in function space, hence performs better than DE and rDE, which purely hinge on the diversity in weight space.

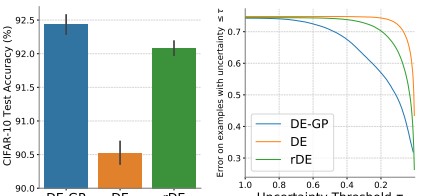

Figure 7: In-distribution test accuracy (Left) and error versus uncertainty plots on the combination CIFAR-10 and SVHN (Right) under weight sharing. (ResNet-20)

## 5.4 CONTEXTUAL BANDIT

Finally, we apply DE-GP to contextual bandit, an important decision-making task where the uncertainty helps to guide exploration. Following (Osband et al., 2016), we use DE-GP to achieve efficient exploration inspired by Thompson sampling. We reuse most of the settings for UCI regression. We leverage the GenRL library to build two contextual bandit problems Covertype and Mushroom (Riquelme et al., 2018). The cumulative reward is depicted in Fig. 8. As desired, DE-GP offers better uncertainty estimates and hence beats the baselines by clear margins. The potential of DE-GP in more reinforcement learning and Bayesian optimization scenarios deserves future investigation.

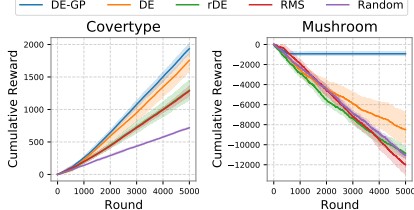

Figure 8: Cumulative reward varies w.r.t. round on Covertype (Left) and Mushroom (Right). Random corresponds to the Uniform algorithm. Summarized over 5 trails.

## 6 CONCLUSION

In this work, we attempt to build a Bayesian refinement of Deep Ensemble by drawing inspiration from the connection between BNNs and GPs. We propose to leverage the ensemble members to specify an adaptive GP approximate posterior, and perform variational inference directly in function space. We inherit some theoretical results from existing works on functional variational inference, and further develop a posterior regularization scheme to conveniently induce prior knowledge on function properties. We demonstrate how to penalize the function complexity based on this scheme and empirically show the necessity of doing so. The whole algorithm can be implemented easily and efficiently. Extensive experiments validate the effectiveness of our method. We hope this work may shed light on the development of better Bayesian deep learning approaches.

ETHICS STATEMENT

This work proposes a Bayesian refinement of Deep Ensemble. Its potential positive impacts in the society are evident: its ability to enable better uncertainty estimation while maintaining predictive performance is crucial in industry, e.g., automatic driving, disease analysis, and financial applications. In this scenarios, the uncertainty estimates could be used to reject uncertain predictions, and raise the requirement of inviting humans into the decision process. As a fundamental research in machine learning, the negative consequences are not obvious. Though in theory any technique can be misused, it is not likely to happen at the current stage.

REPRODUCIBILITY STATEMENT

It is easy to reproduce the experiments given the code in the supplementary material.

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

# A   PENALIZE FUNCTION COMPLEXITY IN MULTI-CLASS CLASSIFICATION

In the multi-class classification scenario where $y \in \{1, 2, ..., C\}$ and $f, g_i : \mathcal{X} \to \mathbb{R}^C$, we use the loss $\ell(f(\boldsymbol{x}), y) = \mathbf{1}_{f(\boldsymbol{x})[y] < \max_{y' \neq y} f(\boldsymbol{x})[y']}$ to measure prediction error where $f(\boldsymbol{x})[j]$ denotes $j$-th coordinate of $f(\boldsymbol{x})$. The distinct difference between this scenario and the binary classification scenario is that in this setting, $\zeta(\boldsymbol{x})$ is a vector-valued function:

$$\zeta(\boldsymbol{x}) = \frac{1}{\sqrt{M}} \sum_{i=1}^{M} \epsilon_i (g_i(\boldsymbol{x}) - m_q(\boldsymbol{x})) + \sqrt{\lambda} \boldsymbol{\epsilon}_0, \tag{18}$$

where $\boldsymbol{\epsilon}_0 \sim \mathcal{N}(\mathbf{0}, \mathbf{I}_C)$ and $\epsilon_i \sim \mathcal{N}(0, 1), i = 1, ..., M$. We then make a mild assumption to simplify the analysis.

**Assumption 1.** *For any $(\boldsymbol{x}, y) \in \mu$, the elements on the diagonal of $k_q(\boldsymbol{x}, \boldsymbol{x})$ have the same value.*

This assumption implies that for any $j, j' \in \{1, ..., C\}$,

$$\frac{1}{M} \sum_{i=1}^{M} (g_i(\boldsymbol{x})[j] - m_q(\boldsymbol{x})[j])^2 + \lambda = \frac{1}{M} \sum_{i=1}^{M} (g_i(\boldsymbol{x})[j'] - m_q(\boldsymbol{x})[j'])^2 + \lambda. \tag{19}$$

I.e.,

$$\sum_{i=1}^{M} (g_i(\boldsymbol{x})[j] - m_q(\boldsymbol{x})[j])^2 = \sum_{i=1}^{M} (g_i(\boldsymbol{x})[j'] - m_q(\boldsymbol{x})[j'])^2. \tag{20}$$

Therefore, $\zeta(\boldsymbol{x})$ possesses the same variance across its output coordinates and becomes a random guess classifier. Based on this, we have $\mathbb{E}_{q(f)} \ell(\zeta(\boldsymbol{x}), y) = \mathbb{E}_{\epsilon_0, ..., \epsilon_M} \ell(\zeta(\boldsymbol{x}), y) = (C-1)/C$. We can then derive a similar conclusion to that in Section 4.3.

**The validity of Assumption 1.** When the data dimension $|\mathcal{X}|$ is high and the number of training data $n$ is finitely large, with zero probability the sampled data $(\boldsymbol{x}, y) \sim \mu$ resides in the training set. Therefore, only the KL divergence term of the fELBO explicitly affects the predictive uncertainty at $\boldsymbol{x}$. Because the NN-GP prior possesses a diagonal structure, it is hence reasonable to make the above assumption.

# B   MORE OF EXPERIMENTS

We provide more experimental details and results in this section. In all experiments, the number of MC samples for estimating the expected log-likelihood (i.e., $U$ in Line 7 of Algorithm 1) is set as 256. Unless otherwise stated, we set the regularization constant $\lambda$ as 0.05 times of the average eigenvalue of the central covariance matrices, and set the weight and bias variance for defining the NN-GP prior kernel at each layer as $2/fan\_in$ and 0.01, where $fan\_in$ is the number of input features, as suggested by He initialization (He et al., 2015).

## B.1   DETAILED SETTINGS

**Illustrative regression.** For the problem on $y = x^3/4 + \epsilon, \epsilon \sim \mathcal{N}(0, 0.1)$, we randomly sample 20 data points from $[-1.5, -0.6] \cup [0.6, 1.5]$. For the problem on $y = x \sin 5x + \epsilon, \epsilon \sim \mathcal{N}(0, 0.2)$, we use 6 data points $\{-0.8, -0.1, 0.02, 0.2, 0.6, 0.8\}$ following (Pearce et al., 2020). For optimizing the DNN basis functions, we use an Adam (Kingma & Ba, 2015) optimizer with 0.003 learning rate. The optimization takes 300 iterations. We set hidden size as 64 for the architecture of the prior NN-GP kernel for speedup. The regularization constant $\lambda$ is set as $1e - 4$ times of the average eigenvalue of the central covariance matrices.

**UCI regression.** We pre-process the UCI data by standard normalization. We set the variance for data noise and the weight variance for the prior kernel following (Pearce et al., 2020). The batch size for stochastic training is 256. We use an Adam optimizer to optimize for 1000 epochs. The learning rate is initialized as 0.01 and decays by 0.99 every 5 epochs.

**Fashion-MNIST classification.** The used architecture is Conv(32, 3, 1)-BN-ReLU-MaxPool(2)-Conv(64, 3, 0)-BN-ReLU-MaxPool(2)-Linear(256)-ReLU-Linear(10), where Conv($x, y, z$) represents

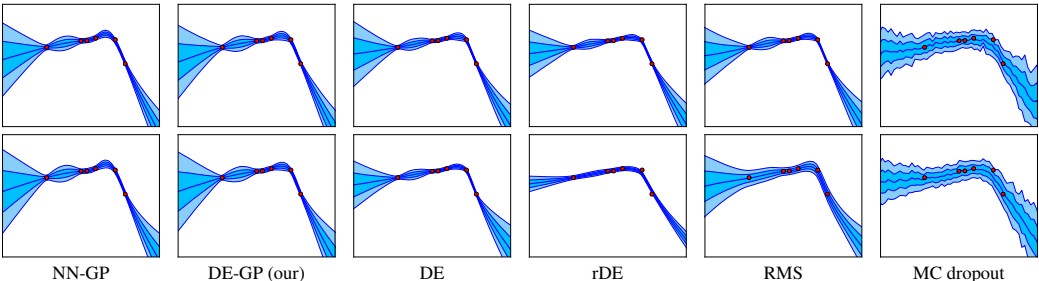

Figure 9: Prediction on toy data from $y = x \sin 5x + \epsilon, \epsilon \sim \mathcal{N}(0, 0.2)$. The two rows correspond to MLPs with 2 hidden layers of 64 units and MLPs with 3 hidden layers of 512 units, respectively. DE-GP provides calibrated uncertainty estimates and is consistently behaved as mode complexity increases.

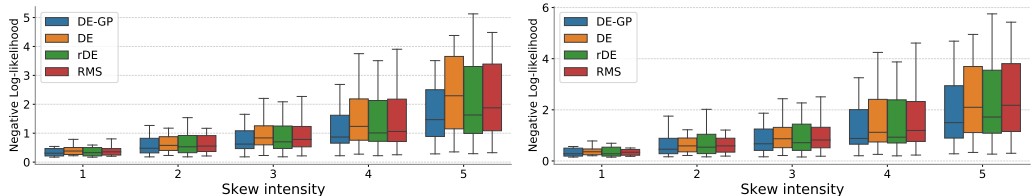

Figure 10: Negative log-likelihood on CIFAR-10 corruptions for models trained with ResNet-20 (Left) or ResNet-56 (Right) architecture. We summarize the results across 19 types of skew in each box.

a 2D convolution with $x$ output channels, kernel size $y$, and padding $z$. The batch size for training data is 64. The batch size for extra measurement points is 0. We use an SGD optimizer to optimize for 24 epochs. The learning rate is initialized as 0.1 and follows a cosine decay schedule. We use an Adam optimizer with $1e - 3$ learning rate to optimize the temperature. We use 1000 MC samples to estimate the posterior predictive and the epistemic uncertainty, because the involved computation is only the cheap softmax transformation on the sampled function values.

**CIFAR-10 classification.** We perform data augmentation including random horizontal flip and random crop. The batch size for training data is 128. The batch size for extra measurement points is 0. We use a SGD optimizer with 0.9 momentum to optimize for 200 epochs. The learning rate is initialized as 0.1 and decays by 0.1 at 100-th and 150-th epochs. We use an Adam optimizer with $1e - 3$ learning rate to optimize the temperature. We use 1000 MC samples to estimate the posterior predictive and the epistemic uncertainty. Suggested by (Ovadia et al., 2019; He et al., 2020), we train models on CIFAR-10, and test them on the combination of CIFAR-10 and SVHN test sets. This is a standard benchmark for evaluating the uncertainty on OOD data.

**Contextual bandit.** We use MLPs with 2 hidden layers of 256 units. The batch size for training data is 512. The batch size for extra measurement points is 0. We update the model (i.e., the agent) for 100 epochs with an Adam optimizer every 50 rounds. We set $\alpha = 1$ and $\beta = 0$ without tuning. DE, rDE, and RMS all randomly choose an ensemble member at per iteration, but our method randomly draws a sample from the GP for decision. This is actually emulating Thompson Sampling and advocated by Bootstrapped DQN (Osband et al., 2016). "Random" baseline corresponds to the Uniform algorithm.

## B.2 MORE RESULTS ON ILLUSTRATIVE REGRESSION

We depict the results on $y = x \sin 5x + \epsilon, \epsilon \sim \mathcal{N}(0, 0.2)$ problem in Fig. 9, which are consistently with those in Fig. 1.

## B.3 MORE RESULTS ON CIFAR-10 CLASSIFICATION

We plot the negative log-likelihood and test accuracy on CIFAR-10 corruptions for models trained with ResNet-20 and ResNet-56 in Fig. 10 and Fig. 11. As shown, DE-GP outperforms the baselines in aspect of negative log-likelihood, but yields similar test accuracy to the baselines. Recapping the results in Fig. 6, DE-GP indeed has improved OOD robustness, but may still face problems in OOD generalization.

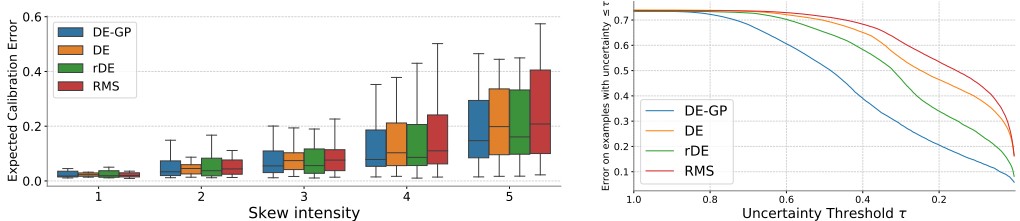

Figure 11: Test accuracy on CIFAR-10 corruptions for models trained with ResNet-20 (Left) or ResNet-56 (Right) architecture. We summarize the results across 19 types of skew in each box.

Figure 12: (Left): Expected Calibration Error on CIFAR-10 corruptions for models trained with ResNet-110 architecture. We summarize the results across 19 types of skew in each box. (Right): Test error versus uncertainty plots for methods trained on CIFAR-10 and tested on both CIFAR-10 and SVHN with ResNet-110 architecture. Ensemble size is fixed as 5 for these experiments.

We then conduct experiments with the deeper ResNet-110 architecture. Due to resource constraint, we use 5 ensemble members. The other settings are roughly the same as those for ResNet-56. The results are offered in Fig. 12, which validate the effectiveness of DE-GP for large networks.

### B.4 RESULTS ON CIFAR-100

We further perform experiments on the more challenging CIFAR-100 benchmark. We present the in-distribution test accuracy of DE-GP as well as the baselines in Table 2. We can see that DE-GP ($\beta = 0.1$) is still on par with rDE. We depict the error versus uncertainty plots on the combination of CIFAR-100 and SVHN test sets in Fig. 13. It is shown that the uncertainty estimates yielded by DE-GP for OOD data are more calibrated than the baselines. We further test the trained methods on CIFAR-100 corruptions (Hendrycks & Dietterich, 2018), and present the comparisons in aspects of test accuracy and NLL in Fig. 14. It is evident that DE-GP reveals lower NLL than the baselines at various levels of skew.

Table 2: Test accuracy comparison on CIFAR-100.

| Architecture | *DE-GP* ($\beta = 0.1$) | *DE* | *rDE* | *RMS* |
|---|---|---|---|---|
| ResNet-20 | 76.59% | 74.14% | **76.81%** | 75.08% |
| ResNet-56 | **79.51%** | 76.46% | 79.21% | 76.77% |

### B.5 ABLATION STUDY ON $\alpha$

We have conducted an ablation study on $\alpha$ (using ResNet-20 on CIFAR-10). The results are presented in Table 3. We can see that DE-GP is not sensitive to the value of $\alpha$. We in practice set $\alpha = 0.01$ in the CIFAR experiments. We did not use a smaller $\alpha$ as it may result in colder posteriors and in turn worse uncertainty estimates.

Table 3: Ablation study on $\alpha$ for DE-GP ($\beta = 0.1$) (using ResNet-20 on CIFAR-10).

| $\alpha$ | 0.1 | 0.05 | 0.01 | 0.005 |
|---|---|---|---|---|
| Accuracy | 94.67±0.09% | 94.66±0.07% | 94.67±0.04% | 94.83±0.10% |

### B.6 ABLATION STUDY ON THE ARCHITECTURE OF PRIOR KERNEL

We perform an ablation study on the architecture for defining the prior MC NN-GP kernel, with the results listed in Table 4. Surprisingly, using the cheap ResNet-20 architecture results in DE-GP with

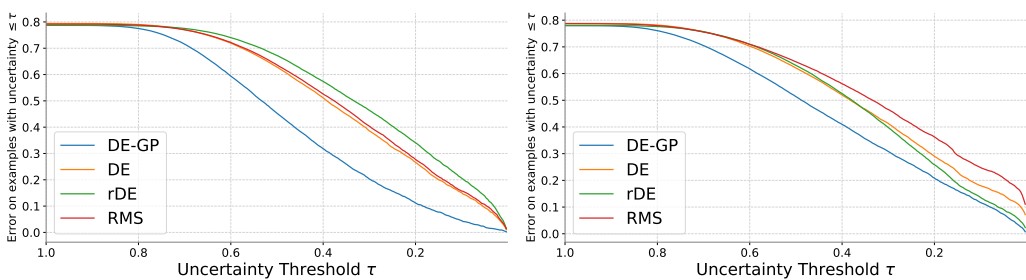

Figure 13: Test error versus uncertainty plots for methods trained on CIFAR-100 and tested on both CIFAR-100 and SVHN with ResNet-20 (Left) or ResNet-56 (Right) architecture.

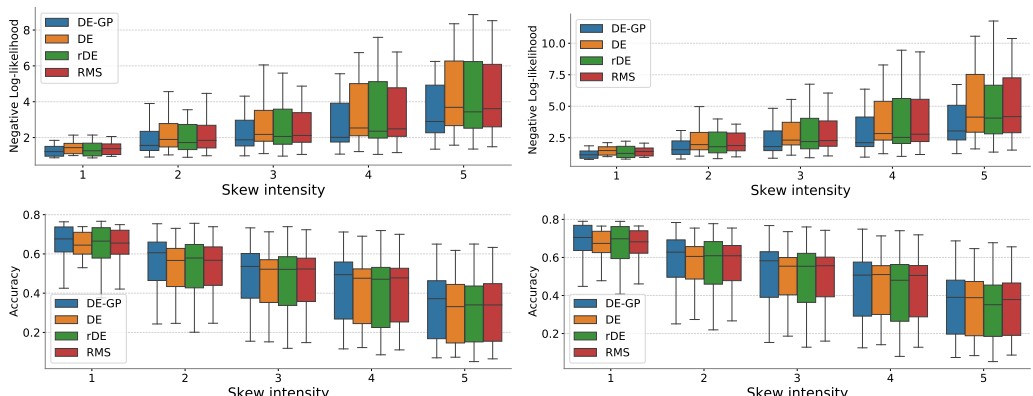

Figure 14: First row: test NLL on CIFAR-100 corruptions for models trained with ResNet-20 (Left) or ResNet-56 (Right) architecture. Second row: test accuracy on CIFAR-100 corruptions for models trained with ResNet-20 (Left) or ResNet-56 (Right) architecture. We summarize the results across 19 types of skew in each box.

better test accuracy. We deduce this is because a deeper prior architecture induces more complex, black-box correlation for the function, which may lead to over-regularization.

Table 4: Ablation study on the architecture of the prior MC NN-GP kernel.

| Prior kernel architecture
DE-GP architecture | ResNet-20 | ResNet-56 | ResNet-110 |
|---|---|---|---|
| ResNet-56 (10 ensemble member) | 95.50% | 95.28% | - |
| ResNet-110 (5 ensemble member) | 95.54% | - | 94.87% |

