# OpenReview forum: "Deep Ensemble as a Gaussian Process Posterior"
_ICLR.cc/2022/Conference — ICLR 2022 Submitted_

### Official Review · Reviewer_hBTZ · 2021-11-01

**Correctness:** 3
**Technical Novelty And Significance:** 2
**Empirical Novelty And Significance:** 2
**Recommendation:** 5
**Confidence:** 4

**Main Review:**

NOVELTY & SIGNIFICANCE

First, I am not sure why this paper is titled deep ensemble as GP posterior while the proposed method instead uses deep ensemble to parameterize the GP posterior. I was under the impression that this paper is about exposing deep ensemble as the natural posterior mean of a GP setup because it aims to provide a Bayesian justification for DE via GP but this is unfortunately not the case -- one can in fact parameterize the GP with any deep network as its kernel but that is not the same as exposing them as inference results of a GP setup, thus providing a Bayesian justification. Given this, the first contribution claimed is kind of moot.

For the same reason as above, this work also appears incremental to me. It is pretty much a special case of GP with deep kernel but perhaps the point here is the DE is a fine choice for parameterization that can improve performance. As for the second contribution that involves the variational inference and regularization scheme, I have to unfortunately say the same thing that these are also too incremental as they are, as a matter of fact, vanilla inheritance from prior work with little to no adaptation:

The variational expression is obviously inherited from (Sun, 2019) as the authors mentioned and it is also known even before that by the prior work of Titsias in AISTATS-09 where an ELBO of GP is derived based on a set of inducing points. The regularization is simply a L2 penalty that was well-rationalized by prior work.

Given the above, I think the key contribution of this work is the empirical investigation of a potential parameterization for a GP with DE, which is fine but marginal given the reported results in Table 1 (i.e. the improvement over DE and rDE is pretty much marginal)

SOUNDNESS

All derivations appear correct to me.

EXPERIMENT

As I mentioned above, the results in Table 1 appear positive but are somewhat marginal (in one setting, rDE is still better). In fact, given the marginal differences, maybe it's worth running for more trials to have more stable estimates -- at this point, every model is within the error bar of almost every other model so it is not clear if the performance is really improved.

On another note, results in the contextual bandit look significant -- perhaps the calibrated uncertainty will show its effect best in an exploit-explore decision making scenario. But I am not sure how the authors set up the uncertainty variance for other non-Bayesian baseline in this experiment? Could the authors clarify this in the rebuttal?

As a more minor note, using bar plot in Fig 7 is kind of inflating the result differences. The difference between DE-GP and rDE is less than 0.5% and we do not know if this is significant without showing the deviation bar -- if it is similar to what was reported in Table 1 then it is not significant.

**Summary Of The Paper:**

This paper presents a GP model with a deep kernel defined in terms of a (finite) deep ensemble (DE). A variational approximation and a regularization scheme were introduced to optimize the GP. The proposed methods were then demonstrated on several benchmark regression/classification datasets.

Here, the claimed contributions are (1) a DE-parameterized GP posterior that provides a Bayesian justification for DE; and (b) a regularization scheme that improves the generalization of the trained GP.

**Summary Of The Review:**

This is an empirical work that aims to demonstrate the uncertainty calibration of DE-GP and its improved performance over a set of DE variants. The results appear somewhat marginal where reported performance of DE-GP is within the error range of other models. More trial runs are necessary to obtain more reliable estimate. Also, I disagree with the claim this work exposes a Bayesian rationalization for DE since this is not what it is doing. Instead, a GP parameterized by DE is introduced and its performance is analyzed empirically.

---

> ### Author Response · Authors · 2021-11-15
> **Thank you for the valuable review**
>
> We thank the reviewers for the insightful comments. We address specific comments below and have incorporated them into the updated version.
>
> ---
>
> **Q1:** Regarding the title and contribution (1).
>
> **A1:** We are sorry that the title may cause some misunderstandings. Your statements are totally correct. However, what you expect – exposing standard deep ensemble as the samples from the posterior of a GP setup – remains challenging to solve. The current works can only expose a refined deep ensemble as posterior samples under certain assumptions, e.g., NTKGP (He et al., 2020) and RMS (Lu & Van Roy, 2017; Osband et al.,2018; Pearce et al., 2020). Our work is slightly different from them, as you understand, we also *build a Bayesian refinement of deep ensemble*, but it is in the form of GP, and we perform variational inference for training. We think the two directions are both tenable. We thank the reviewer for pointing out this, and we have revised the whole paper accordingly.
>
> By the way, the reviewer's understanding that "DE is a fine choice for parameterization that can improve performance" is also correct. But we expect to clarify that we are not dedicated to building a powerful deep kernel, but aim to *build a Bayesian refinement of deep ensemble*. We hope that the reviewer can realize the difference between these two things and approve the significance of the thing that we focus on.
>
> ---
>
> **Q2:** Regarding contribution (2).
>
> **A2:** Yes, the technical contribution of the function-space variational inference part is limited and we have already acknowledged this in Section 4.2. However, we respectfully disagree with the reviewer on the regularization. In fact, the L2 penalty on weights rarely appears in the previous function-space Bayesian inference works, for example, it doesn't appear in the function-space particle optimization variational inference (Wang et al., 2019). The reason is that in function-space Bayesian inference, the regularization is applied on functions (like the kernel ridge regression). And people may think that introducing an extra weight-space penalty may lead to an unprincipled posterior inference. However, without the weight-space penalty, the bases specifying the deep kernels would suffer from high complexity (see Figure 2), and hence result in poor posterior predictive performance (see Table 1). We resort to the *posterior regularization* paradigm (Ganchev et al., 2010; Zhu et al., 2014) to make the weight-space regularization tenable, and to make the whole framework applicable to modern architectures like ResNets.
>
> ---
>
> **Q3:** Results in Table 1.
>
> **A3:** Thanks for the suggestion. We have run each experiment 5 more times (hence totally 8 trials per experiment) and updated Table 1. We notice that the accuracy of DE-GP ($\beta=0.1$) is still on par with rDE, consistent with our previous results. As we originally claimed, DE-GP brings better uncertainty estimates without compromising accuracy. We think our experiment section can support this claim.
>
> ---
>
> **Q4:** How to set up the uncertainty variance for other non-Bayesian baselines in the contextual bandit experiments?
>
> **A4:** We are sorry for the missing details. We kindly point out that DE, rDE, and RMS all randomly choose an ensemble member at per iteration (as they regard the ensemble members as a set of approximate samples from the posterior), but our method randomly draws a sample from the GP defined in Eq (5) for decision. "Random" baseline corresponds to the Uniform algorithm.
>
> ---
>
> **Q5:** Show the deviation bar in Figure 7.
>
> **A5:** Thanks for the suggestion. We have run the experiments 5 more times (hence totally 6 trials per experiment) and report the mean and std in the updated Figure 7. It is clear that DE-GP is better than rDE.

---

### Official Review · Reviewer_Bycy · 2021-11-01

**Correctness:** 3
**Technical Novelty And Significance:** 2
**Empirical Novelty And Significance:** 2
**Recommendation:** 5
**Confidence:** 3

**Main Review:**

## Contributions & Weaknesses
The paper shows empirically that DE-GP achieves somewhat better RMSE and accuracy compared to most baselines across a variety of synthetic, regression, and low dimensional vision tasks. However, these gains do not appear significant or well motivated. It is not clear what pain points of deep ensembles "not being Bayesian'' are being  addressed and what concrete value the DE-GP method brings the other forms of approximate inference against which the method is compared. Like the other Bayesian methods of approximate inference, DEs have empirically shown closer approximations to the predictive posterior distribution, so it is unclear what is meant when seeking "a more proper Bayesian exposition for DE", and whether this distinction of DEs being "non-Bayesian" is necessary. If the goal is higher fidelity Bayesian inference, this need not be achieved by incorporating a GP. It makes sense to want to improve the generality of approximate models across various tasks, but significance could be better emphasized by specifying an argument against other state of the art methods, either theoretically or empirically, or by highlighting significant advantages / capabilities of the proposed method.

Theoretically, there is little theoretical novelty in setting up the variational inference optimizing framework and defining the mean and covariance of basis functions, as these seem to be trivial adaptations of prior works. In terms of empirical results, there does not seem to be significant differences between NN-GP, DE-GP, and DE in most cases. Figures 1 and 9 are also not super informative in terms of emphasizing differences or giving insight into why or how DE-GP is competent at exploring and averaging uncertainty under a single mode, as noted that ensembles may not always select the best points from each mode. It is also difficult to justify better calibration purely from the visual.

## Correctness & Clarity
The paper is well written overall. The proofs and math are clear and in line with prior works (e.g. Theorem 1 of [3] is applied for developing the objective in section 3.2). It could be better specified which parts of the pipeline are original work to emphasise contribution, as it currently only states which parts are borrowed.

In various parts of the paper, the method is motivated by the fact that DEs "lack a proper Bayesian justification". Could this be explained in more specificity? This is confusing and the paper does not clarify how and why certain methods are considered Bayesian or not. It is stated that the objective of the paper is to make DEs more Bayesian, but also noted that doing so via DE-GP may be at the cost of "trad[ing]-off between theoretical soundness and flexibility in practical usage". Flexibility should be clarified and justified for the DE-GP, otherwise, it cannot be used as evidence against employing DE in Bayesian inference problems.

## Relation to prior work
Generally well-covered relevant works session, cites major works that would be good additional baselines. Methods are developed from past works on basis functions and GPs.

## Additional feedback (comments, suggestions for improvement)
- Section 3.1 claims that the chosen linear, matrix value kernel is comparable to recent low rank approximations to original kernel matrices. The authors should provide comparisons with methods like Batch Ensemble [1], Rank-1 BNNs [2], etc. to demonstrate this claim.
- It would be nice to do an ablation to better motivate why VI is necessary and important to consider in Bayesian inference. There should be better characterizations of local approximations including Laplace approximation as well as global behaviours with MCMC, HMC or SGLD. Additionally, negative log likelihood appears better than others, but there is no comparison with NN-GP on CIFAR-10 for example. It would also be good to vary the ensemble size and compare it with NN-GP.
- As uncertainty quantification is a major focus of most BNN / GP papers, additional methods and baselines should be included in the comparisons: mean field V I (motivating variational inference), AugMix (insights into the effects of data augmentations), ResNet architectures with and without batch normalization (analyze inductive biases contributed by architectures and generally how far off from most performant models).
- The paper claims that DE-GP endows high complexity in function space. Can you show why this matters? For example, it would be more exciting and interesting to use a mixture of Gaussians as basis and show that you can / cannot achieve non-Gaussian and/or multimodal posteriors.
- Further uncertainty and robustness analysis can be justified. It could be informative to measure the quality of robustness through additional experiments on tasks that encourage robustness to novel modalities (e.g. textures in vision tasks, corruptions / covariate shapes etc.)
- Since the GP has a linear kernel, the method is identical to a Bayesian linear layer after a mean-only layernorm stage. Describing this model as an ensemble is misleading as well, since the components are jointly trained in a single objective as opposed to separately.

## Questions for the authors
1. More emphasis should be placed to detail the problem the paper is hoping to solve. Is it with regards to inference, prior selection, or something else more specifically?
2. How do the computational costs of training DE-GPs compare to baseline and recent methods?
3. How is expressivity of the posterior affected by the choice of prior process? Does a "good" choice of prior also allow for better tradeoff between cost and uncertainty estimation?
4. Do you think a choice of non-stationary kernel as prior processes could induce better inductive biases and hence be conducive to stronger OOD detection?
5. Have you tried experimenting with different ways of doing model aggregation in the ensemble framework?

### References
[1] https://arxiv.org/abs/2002.06715
[2] https://arxiv.org/pdf/2005.07186.pdf
[3] https://arxiv.org/pdf/1903.05779.pdf
[4] https://arxiv.org/pdf/1711.00165.pdf


**Summary Of The Paper:**

The authors propose using Deep Ensemble (DE) as basis functions to train a Gaussian Process (GP) with the primary motivation of making DEs more Bayesian. The Evidence Lower BOund (ELBO) being maximized is now optimized via variational inference (VI) over the functional space of bases (i.e. fELBO from [3] Sun et al.) defining the mean and covariance of a GP similar to NN-GP [4]. The paper evaluates the proposed method on standard UCI regression and small-scale MNIST / CIFAR10 image classification tasks in terms of accuracy, likelihood, and uncertainty estimation.

**Summary Of The Review:**

Overall this paper demonstrates theoretical soundness and evaluates against standard benchmarks in the space of models with uncertainty estimation. However, it lacks some originality as well as comparisons with additional baselines and ablations to better separate its significance and reason for performing better / worse. There is also no clear indication of the problem being addressed and why being Bayesian matters.

---

> ### Author Response · Authors · 2021-11-15
> **Thank you for the valuable review (Part 3/3)**
>
> **Q4:** Questions for the authors
>
> **A4:**
>
> - *What is the problem the paper is hoping to solve?*
>
>   We aim at developing a Bayesian refinement for deep ensemble.
>
> - *Computational costs comparison.*
>
>   A direct comparison is provided in Figure 1. We have also discussed in Section 4.5, and the conclusion is that "DE-GP is only marginally slower than DE in training".
>
> - *Effects of the prior.*
>
>   We agree with the reviewer that a "good" prior would lead to a posterior with high quality or specific properties that we want. Currently, we are using the (MC) NN-GP priors, which correspond to the conventionally used weight-space Gaussian priors. Namely, we follow the common practice for defining priors in (Bayesian) deep learning. Similar setups can be found in some closely related works like f-POVI (Wang et al., 2019). Searching over a space to find a better prior is a viable way to boost our approach.
>
> - *"Do you think a choice of non-stationary kernel as prior processes could induce better inductive biases and hence be conducive to stronger OOD detection?"*
>
>   We don't know. Currently, the used (MC) NN-GP kernel is actually a non-stationary kernel. We think the thing of more importance is the architecture specifying the kernel instead of whether it is stationary or not.
>
> - Have you tried different ways of doing model aggregation in the ensemble framework?
>
>   No. We assemble the models according to the *posterior predictive* distribution.

---

> > ### Comment · Reviewer_Bycy · 2021-12-01
> > **Acknowledgement of response**
> >
> > Thank you for responding to my various questions, I am satisfied with the answers. I am generally in agreement with the other reviewers that this work is incremental in the theoretical sense, and given this, could have also shown a more in depth empirical analysis as compared to other works in the Bayesian domain. I am also in agreement that making deep ensembles "more Bayesian" isn't clearly motivated. I maintain my score.

---

> ### Author Response · Authors · 2021-11-15
> **Thank you for the valuable review (Part 2/3)**
>
>
> **Q3:** Comments, suggestions for improvement
>
> **A3:**
>
> - *The claim that "the chosen linear, matrix value kernel is comparable to recent low-rank approximations to original kernel matrices".*
>
>   We respectfully point out that this is a misunderstanding. We didn't claim that *our kernel is comparable to recent low-rank approximations to original kernel matrices*. We just want to express that modeling low-rank variations is a fashion in Bayesian deep learning, and use this argument to support our modeling choice. Besides, BatchEnsemble and Rank-1 BNNs are not "low-rank approximations to original kernel matrices" -- these two works perform inference in the weight space, so there is no kernel involved. What's more, the baselines of concern in our paper include the DE and rDE. They are generally stronger than BatchEnsemble and Rank-1 BNNs (see their papers).
>
> - *Why is only VI considered to refine deep ensembles? Can we use other Bayesian inference methods?*
>
>   We are sorry that we cannot provide such an ablation study. The goal of this paper is to find *one possible* approach to make deep ensemble possess a Bayesian justification. Developing Laplace approximation or MCMC methods to achieve the same goal would give rise to another paper. Besides, we doubt that the Laplace approximation would not yield a better approximation than VI, which has been extensively revealed in the past research. Combining HMC/SGLD and deep ensemble seems to be a promising direction, but we are not clear how to achieve so at least at the current stage. Could you provide more details on it?
>
> - *More comparisons with NN-GP.*
>
>   As stated in Section 2.2, we can hardly estimate the exact NN-GP posteriors when facing classification problems, large data (Shi et al., 2019), and contemporary DNN architectures (Novak et al., 2018), e.g., ConvNets with max-pooling and BNs.
>
> - *The results vary w.r.t. the ensemble size.*
>
>   We provide such results on FMNIST in Figure 4. We will incorporate the results on CIFAR-10 in the next version.
>
> - *Compare to mean-field VI, AugMix, ResNets w/o BNs.*
>
>   Thanks for the advice. However, we clarify that we didn't compare to mean-field VI because DE and rDE are much stronger baselines than mean-field VI. Besides, we motivate function-space VI by regarding DE as an approximate function-space posterior, so we don't need to compare to mean-field VI to motivate our approach. Currently, we train models using standard data augmentation (random crop/flip) and ResNet w/ BNs. Comparing to models trained with stronger data augmentation or architectures w/o BNs is possible, and we will try to add these results in the final version.
>
> - *"DE-GP endows high complexity in function space", what are the causes and effects?*
>
>   As the DNNs are highly non-convex, the L2 regularization on function values doesn't necessarily lead to L2 regularization on weights. So, the KL in the fELBO is less effective in controlling the capacity of the parameterized models (see Figure 2). Then, the approximate posteriors become high-complexity and deliver poor generalization performance (see Table 1). Thereby, we propose to leverage the *posterior regularization* paradigm (Ganchev et al., 2010; Zhu et al., 2014) to control the complexity of DE-GP, and in turn to improve generalization.
>
> - *"It would be more exciting and interesting to use a mixture of Gaussians as basis and show that you can / cannot achieve non-Gaussian and/or multimodal posteriors."*
>
>   We appreciate the suggestion. But, we cannot understand the detailed meaning. In our opinion, the basis is a function, so why do you say "use a mixture of Gaussians as basis"? Can you elaborate more on this?
>
> - *Additional experiments on tasks that encourage robustness to novel modalities.*
>
>   Thanks for the constructive advice. We have included the experiments on CIFAR-10 corruptions in the paper. We would like to try on the settings you mentioned.
>
> - *The method is identical to a Bayesian linear layer.*
>
>   We are sorry for some misleading descriptions in the paper. But, actually, our method is *not* identical to a Bayesian linear layer. We conjecture that you regard the DE-GP kernel as a prior kernel and hence obtain such a misunderstanding. Another possibility is that you think our DE-GP is similar to deep kernel learning (Wilson et al., 2016), but actually, DE-GP is significantly different from it. Recall that we define the deep ensemble as a variational and apply function-space VI for learning.
>
> - *"Describing this model as an ensemble is misleading."*
>
>   Thanks for pointing that out. Yes, we in fact call our model a Gaussian process, or more specifically DE-GP.

---

> ### Author Response · Authors · 2021-11-15
> **Thank you for the valuable review (Part 1/3)**
>
> Thank you so much for your efforts in assessing our work. We appreciate the valuable suggestions and have revised the paper accordingly. We now answer your questions in detail.
>
> ---
>
> **Q1:** Regarding contributions & weaknesses.
>
> **A1:**
> - *Why do deep ensembles "not being Bayesian"? What are the pain points of deep ensembles?*
>
>   In a typical Bayesian approach, we are interested in finding the *posterior* distribution of some random variables given the *prior belief* and the *data likelihood*. The benefits of a Bayesian approach naturally include the Bayesian uncertainty. By this definition, it is hard to interpret DE as a Bayesian approach. The “uncertainty” yielded by DE totally stems from the randomness in DNNs initialization and SGD, and hence it is just some kind of stochasticity instead of the Bayesian uncertainty.
>
>    The main pain point of DE is then that there is no guarantee that the uncertainty estimates given by DE are reliable (see Figure 1: sometimes DE yields good predictive distributions, while sometimes yields bad ones). That is why many researchers are still devoted to providing more proper Bayesian justification for deep ensemble (Lu & Van Roy, 2017; Osband et al., 2018; Pearce et al., 2020; He et al., 2020).
>
>   So, our contribution is that we leverage function-space variational inference to *explicitly* regularize DE, which guarantees the quality of the approximate posterior predictive and the uncertainty estimates.
>
>   We have made these clear in the updated paper.
>
> - *Why is DE-GP superior over the baselines that also make deep ensemble Bayesian?*
>
>   As we discussed in related works section, these baselines routinely make non-trivial assumptions such as *linear data likelihood* or *infinitely wide model*. These cannot be fulfilled in practice. Yet, DE-GP does not rely on these limited assumptions.
>
> - *"It makes sense to want to improve the generality of approximate models across various tasks, but significance could be better emphasized by specifying an argument against other state-of-the-art methods."*
>
>   Thanks for the advice. In the revised paper, we highlight that DE-GP enables us to train deep ensemble under Bayesian principle without restrictive assumptions as made by the related works.
>
> - *The theoretical novelty is little.*
>
>   We agree with the reviewer. Yet, we clarify that this paper does not aim at developing novel theories or techniques, but to provide a suitable justification for deep ensemble. This point is appreciated by Reviewers 5XQ4 and  zZzc.
>
> - *"There does not seem to be significant differences between NN-GP, DE-GP, and DE in most cases."*
>
>   We clarify that the comparisons in terms  of uncertainty on FMNIST (see Figure 4), CIFAR-10 (see Figure 5, 6), and the contextual bandit (see Figure 8) have proved that DE-GP is better than DE and rDE across settings. NN-GP cannot even be applied in these scenarios due to the difficulties of kernel computation and analytical inference. Reviewers uL2a, 5XQ4 and zZzc all approve our experiments.
>
> - *Figures 1 and 9 are not super informative in terms of emphasizing differences or giving insight into ...*
>
>   Thanks for the advice. Figures 1 and 9 are now just for conceptually demonstrating the unreliability issue of deep ensemble uncertainty and the effectiveness of DE-GP. They are in line with the plots in some related works (Sun et al., 2019). We have not fully understood your advice and welcome further clarification on this.
>
> ---
>
> **Q2:** Regarding correctness & clarity.
>
> **A2:**
>
> - *Make clear which parts of the pipeline are original work to emphasize contribution.*
>
>   Thanks for the advice. We have revised the paper (especially the Related Works and Methodology sections) carefully to emphasize our contributions while avoiding over-claim. Feel free to inform us if there is still something improper.
>
> - *Trade-off between theoretical soundness and flexibility in practical usage.*
>
>   Thanks! As we pointed out in the Discussion section, DE-GP is likely to be less flexible than DE because all the basis functions in DE-GP need to be updated *simultaneously* while DE doesn't entail doing so. So, there is indeed a trade-off, and we would like to build more flexible approaches beyond DE-GP.

---

### Official Review · Reviewer_zZzc · 2021-11-02

**Correctness:** 3
**Technical Novelty And Significance:** 2
**Empirical Novelty And Significance:** Not applicable
**Recommendation:** 5
**Confidence:** 3

**Main Review:**

### Strengths

-	The paper is very well-structured overall and a pleasure to read. I particularly enjoyed the sections describing the contributions, as well as the brief discussion in Section 3.5 that summarises some of the key properties of the proposed method.
-	The dilemma on whether ensemble methods are just as effective as fully Bayesian methods has been the subject of several papers and discussion panels in recent years - the insights and contributions presented in this paper are thus timely and relevant, and should be of interest to the community.
-	The experiments section is very thorough, and covers a vast set of problems ranging from synthetic examples for visual exposition of the model’s features, regression datasets, and classification problems. The paper also contains an additional experiment on a contextual bandit problem that I particularly appreciated, as it shows how the improved uncertainty estimates returned by this method can be beneficial when used in iterative decision making tasks. The experimental set-ups are well-explained in each case, and I appreciated that the results were all computed from scratch using implementations of existing methods (as opposed to simply copying in results from other papers that could possibly have been computed over slightly different dataset folds and experimental settings).

### Weaknesses

-	Although I previously praised the paper for being well-organised, there are several instances where the writing can be improved. There are quite a few typos and grammatical errors scattered throughout the paper which could easily have been fixed with a proper read-through before submission. The introduction is particularly disappointing and I would recommend rewriting. There are also a few phrases which are ambiguous to a point and I would also reconsider. For example, the reference to ‘graceful BNNs’ in the conclusion is quite odd.
-	The related work section appearing towards the end of the paper does a good job in exhaustively listing the various papers working on areas related to this topic, but offers very little insight into their actual contributions. Perhaps it may be more suitable to place this section earlier in the paper such that any contributions can be more clearly referenced against earlier work?
-	While interesting, the principal contributions of the paper are fairly incremental, and the authors themselves concede leveraging insights ‘inherited’ from other works as the foundation for some of the key contributions. Even so, this does not dent my overall opinion of the paper.
-	The references need to be properly cleaned up - journal versions and conference proceedings should be cited instead of Arxiv editions of certain papers, while words such as Gaussian need to consistently appear as capitalized in paper titles.


**Summary Of The Paper:**

The motivation for using fully Bayesian methods over ensemble methods has been a contentious topic in recent years - while ensemble methods are prized for their relative simplicity, they lack the theoretical framework that grounds fully Bayesian approaches. In this work, the authors propose an interpretation of Deep Ensemble models (DE) under a variational Bayesian framework. In particular, the authors demonstrate how VI can be carried out in the function space, and leverage posterior regularisation on functions to incorporate prior knowledge into the model architecture. The benefits of the proposed approach are verified via an extensive evaluation covering a variety of different problem settings, whereby it appears that the DE-GP consistently yields predictions having superior uncertainty calibration, and without compromising on predictive accuracy.

**Summary Of The Review:**

The paper tackles an interesting subject in an intuitive manner, and the extensive experimental evaluation showcases the performance of the model in several different settings. On the downside however, some of the writing needs to be heavily improved, while the connections to related work should also be reconsidered to further highlight the original contributions of this work.

---

> ### Author Response · Authors · 2021-11-15
> **Thank you for the encouraging review!**
>
> We thank the reviewer for the encouraging review. We appreciate the acknowledgment of our paper's strengths in aspects of writing, insights, contributions, and experiments. We also appreciate the constructive suggestions for improving the presentation. We address specific comments below:
>
> ---
>
> **Q1:** Typos, grammatical errors, the introduction section, etc.
>
> **A1:** Thanks a lot for the suggestions. We have revised the paper carefully to reduce the typos and grammatical errors. We have also re-written the introduction section. Please let us know if you have further comments.
>
> ---
>
> **Q2:** Regarding related work.
>
> **A2:** Thanks for the suggestion. We follow the advice and place the "related work" section earlier in the paper to better clarify our contributions and the contributions of the related work.
>
>
> ---
>
> **Q3:** References.
>
> **A3:** We appreciate the kind suggestion. We have polished the references in the updated paper.

---

> > ### Comment · Reviewer_zZzc · 2021-11-22
> > **Acknowledgement of Rebuttal**
> >
> > Thank you very much for your rebuttal, and for replying to all reviews. I have read all the provided feedback and your corresponding responses, and will engage with the other reviewers in the ensuing discussion.

---

> > > ### Author Response · Authors · 2021-11-29
> > > **Thanks!**
> > >
> > > Dear reviewer, thanks for your time and efforts in reviewing our paper.
> > >
> > > We are glad to see the acknowledgment that *the insights and contributions presented in this paper are thus timely and relevant, and should be of interest to the community.*
> > >
> > > Given the improved presentation and empirical studies, we sincerely hope the reviewer can consider increasing the score.
> > >
> > > Best,
> > > The authors

---

### Official Review · Reviewer_5XQ4 · 2021-11-03

**Correctness:** 4
**Technical Novelty And Significance:** 4
**Empirical Novelty And Significance:** 3
**Recommendation:** 8
**Confidence:** 4

**Main Review:**

Strengths:
- The proposed methods is elegant and provides Bayesian justification on top of DE.
- Extensive experiment evaluation.
- DE-GP shows good performance compared to DE.

Some questions:
- Algorithm: How to choose $\lambda$?
- Algorithm Line 7: what is U?
- why is "the diversity on the points far away from the training data is further fostered"?
- why not search over $\beta$ since it seems to change for different experiments?
- How sensitive is the performance with the changes of the hyperparameters $\alpha, \beta$?

Minor comments:
- Page 5: Require revising "given the fact that a voting is incorrect when at least one of the individuals makes a mistake"


**Summary Of The Paper:**

The paper presents a novel perspective on deep ensembles (DE) that aims at providing a Bayesian justification to DE. The proposed approach builds a Gaussian process (GP) using the ensemble members and performs variational inference in the functional space. Moreover, the paper introduces a regularization method that works directly in the function space. The proposed algorithm DE-GP shows better results compared to DE an other well known Bayesian deep learning methods.


**Summary Of The Review:**

Overall, I find the contributions presented in the paper novel and important to the Bayesian deep learning researchers to know about. Such contributions open the door for more works adapting Bayesian deep learning methods.

---

> ### Author Response · Authors · 2021-11-15
> **Thank you for the supportive review!**
>
> We thank the reviewer for the positive review, which truly reflects many of the essential contributions of our work. Regarding the specific comments, we answer as follows:
>
> ---
>
> **Q1:** Algorithm: How to choose $\lambda$?
>
> **A1:** We set $\lambda$ as 0.05 times of the average eigenvalue of the central covariance matrices $\frac{1}{M}\sum_{i=1}^M (g_i - m_q^{\tilde{X}_s})(g_i - m_q^{\tilde{X}_s})^\top$ following some practice in the community (Park et al., 2020), as stated in Appendix B. The main function of $\lambda$ is to avoid singularity.
>
> > Towards NNGP-guided Neural Architecture Search, Park et al., 2020.
>
> ---
>
> **Q2:** Algorithm Line 7: what is $U$?
>
> **A2:** We are sorry for the missing details for $U$. In the initial paper, we explained in Appendix B that $U$ is the number of MC samples for estimating the expected log-likelihood (i.e., the first part of the RHS of Eq (14)). We have fixed this issue in the updated paper.
>
> ---
>
> **Q3:** Why is "the diversity on the points far away from the training data is further fostered"?
>
> **A3:** We are sorry that the argument is not suitably explained. On one hand, the prior, e.g., a standard GP/MC NN-GP, usually has a non-degraded variance, and hence minimizing the KL between the DE-GP variational and the prior will encourage the DE-GP variational to have non-degraded variance as well, which means that there is function-space diversity among the ensemble members. On the other hand, the other part in the fELBO -- the expected log-likelihood, would enforce each ensemble member to yield the same, correct outcomes for the training data, thus the diversity among the ensemble members on the training data is suppressed. Thereby, we made that claim. Figure 1 provides an illustration for this. We have revised the paper to make this clearer.
>
> ---
>
> **Q4:** Why not search over $\beta$ since it seems to change for different experiments?
>
> **A4:** Thanks for the advice. Currently, we set $\beta=0$ when the model capacity is low (e.g., MLPs or shallow ConvNets), and set $\beta$ according to *the regularization intensity on weight in rDE* when using large models like ResNets. We have not explicitly tuned $\beta$, and doing so may indeed improve the results.
>
> ---
>
> **Q5:** How sensitive is the performance with the changes of the hyperparameters $\alpha, \beta$?
>
> **A5:** As stated in the last reply, we can set $\beta$ according to some principles to avoid tuning.
> Since that we fix some critical hyper-parameters for specifying the NN-GP prior (e.g., the prior variance on weights and biases), it is then necessary to tune $\alpha$ over a range of possible values to better trade-off between the data fitting and the priori inductive bias. Following the reviewer’s advice, we have conducted an ablation study on $\alpha$ (using ResNet-20 on CIFAR-10), and here are the results:
>
> | $\alpha$ | 0.1 | 0.05 | 0.01 | 0.005 |
> | -------- | -------- | -------- | -------- |-------- |
> | Accuracy |   94.67±0.09% |   94.66±0.07%   | 94.67±0.04%  |      94.83±0.10% |
>
> We found that DE-GP is not sensitive to the value of $\alpha$.  We in practice set $\alpha=0.01$ in the CIFAR experiments. We did not use a smaller $\alpha$ as it may result in colder posterior and in turn worse uncertainty estimates.
>
>
>
> ---
>
> **Q6:** Require revising "given the fact that a voting is incorrect when at least one of the individuals makes a mistake".
>
> **A6:** Thanks for the advice. We have revised it in the updated paper.

---

> > ### Comment · Reviewer_5XQ4 · 2021-11-22
> > **Comment on Rebuttal**
> >
> > I would like to thank the authors for addressing all my comments. I am satisfied with
> >  the provided answers.

---

### Official Review · Reviewer_uL2a · 2021-11-03

**Correctness:** 4
**Technical Novelty And Significance:** 3
**Empirical Novelty And Significance:** 3
**Recommendation:** 5
**Confidence:** 4

**Main Review:**

## Methodology

The method is, as far as I understand, the following:
1. We define an NN-GP Gaussian process prior.
2. We define a variational Gaussian process family, where the mean and covaraince are parameterized through the deep ensemble.
3. We introduce a modified ELBO, where (a) we use a tunable weight for the KL, (b) we have an extra regularization term on the weights of the ensemble members and (3) the KL is calculated using the marginal distributions on the train set and additional randomly sampled inputs.

The authors claim that this is "theoretically sound" and "principled" "Bayesian" method. However, essentially, we are using a variational GP model. If we ignore the $||w_i||_2^2$ term in EQ 13, then the optimal solution of the ELBO optimization problem is recovering the NN-GP model. However, the method seems to work better than the NN-GP in practice, and, if the goal was to approximate NNGP, why wouldn't we just use NNGP? Is the idea that the specific family in which we approximate the posterior of the NN-GP the key component here, and what we want is the closest possible deep ensemble to the NNGP posterior? The motivation isn't fully clear to me.

## Section 3.3

I think section 3.3 is very confusing. Essesntially, the whole purpose of this section is to introduce a regularizer on the norm of the weights of the ensemble components, $||w_i||_2^2$. However the authors introduce it by first talking about constrained VI, than saying that our constraint will be that the accuracy on the train set should be good, then bounding the accuracy on train with the accuracy of ensemble components, and then saying that the accuracy of ensemble components is bounded by the norm of the weights.

I have a few issues with this section:
1. I think your constrained in the constrained VI should not depend on the data. Restricting your model class to only the functions that get good accuracy on your train set doesn't seem like a valid Bayesian procedure. Accuracy on train should be a part of your likelihood, not constraint on the model class.
2. I don't think there are bounds on DNN performance that are based just on the $L_2$  norm of the weights.
3. The regularization on the weights is a simple-enough idea, I think it would be much more clear if you just said that you are adding an extra weight decay term without drawing parallels to constrained VI and generalization bounds.

## Making deep ensembles Bayesian

One of the goals that the authors repeat multiple times in the paper is "make deep ensembles a Bayesian method". In light of [this recent blog-post](https://cims.nyu.edu/~andrewgw/deepensembles/) I would like to ask the authors what exactly constitutes a Bayesian method, and why the proposed procedure is "more Bayesian" than the original deep ensembles?

The recent paper [1] also proposes a way of making deep ensembles "more Bayesian", so I think it should be discussed in the paper.

## Experimental results

I think the experimental results are the main strength of the paper. It seems like the method proposed by the authors performs well in a few practical settings, including CIFAR-10 and CIFAR-10-C. However, the performance is only marginally better than regularized deep ensembles on the in-distribution CIFAR-10 benchmarks.

I think the experiments could still be made stronger by including other realistic datasets, perhaps CIFAR-100.

## Questions

- Do you train the ensemble from scratch using Algorithm 1, or do you pretrain the ensemble components first?


[1] Repulsive Deep Ensembles are Bayesian
Francesco D'Angelo, Vincent Fortuin

**Summary Of The Paper:**

The paper develops a deep ensemble-based Gaussian process model and a variational inference procedure to train it. As a result, the authors obtain a method for improved uncertainty and predictions with deep ensembles.

**Summary Of The Review:**

The paper presents a method with promising empirical results. However, the authors frame the main contribution of the paper as making a principled a theoretically sound version of deep ensembles. I do not fully agree with this interpretation given the details of the method. So, I vote for a weak reject.

---

> ### Author Response · Authors · 2021-11-15
> **Thank you for the valuable review (Part 2/2)**
>
> **Q4:** Experiment results.
>
> **A4:** Yes, the test accuracy on CIFAR-10 of DE-GP is only marginally better than that of rDE, but the uncertainty estimates provided by DE-GP are better than those of rDE (see the error versus uncertainty plots and the ECE comparisons on CIFAR-10 corruptions).
> Following the reviewer's advice, we've added the results on CIFAR-100 in Appendix B.4. We also present the results here:
>
>
>
> |Architecture | DE-GP ($\beta=0.1$) | DE | rDE | RMS|
> | -------- | -------- | -------- |-------- | -------- |
> | ResNet-20  | 76.59% | 74.14% | **76.81**% | 75.08% |
> | ResNet-56  | **79.51**% | 76.46% | 79.21% | 76.77% |
>
>
> We can see that DE-GP is still on par with rDE in terms of accuracy. We will add more results on uncertainty estimation in the next version.
>
> ---
>
> **Q5:** Do you train the ensemble from scratch using Algorithm 1?
>
> **A5:** Yes. We trained the ensemble from scratch.

---

> > ### Comment · Reviewer_uL2a · 2021-11-18
> > **Thank you for the clarifications**
> >
> > Dear authors, thank you for your clarifications! I am still not sure about parts of the response:
> >
> > **Q1.** I am a bit confused about the difference between NN-GP and MC NN-GP. From the paper it seems like your MC NN-GP is still a Gaussian process, while a finitely wide Bayesian neural network with a Gaussian prior over the parameters is not in general a Gaussian process. Could you please clarify what exactly is the prior over functions in this case?
> >
> > **Q2.** I still think that the preserntation of the weight decay term is just misleading.
> >
> > > As a result, the first issue raised by the reviewer does not exist. We indeed guarantee the accuracy of the training data by the likelihood, but to make the trained model more generalizable by the constraint.
> >
> > Even if you use the likelihood to bound the true risk, you are still using the data likelihood as your constraint. However, this indeed isn't a real issue because you never really use this constraint and replace the data likelihood with the norm of the weights... Why not just say that your constraint is directly constraining the norm of the weights without going through multiple (imho unrelated) logical steps?
> >
> > > We also clarify that there are extensive empirical and theoretical works showing that the generalization error of DNNs can be reduced by norm-based model capacity control (Neyshabur et al., 2015; 2017; Bartlett et al., 2017; Jiang et al., 2019).
> >
> > However, these bounds are not *just* the norm of the weights, and these bounds often involve norms different from L2. I understand that bounding the norm has been linked to generalization, but just replacing the generalization error with the L2-norm of the weights is not a "principled" transition.
> >
> > > ... directly saying we add an extra weight-space regularization to the function-space variational inference framework would make the readers think that our approximate inference is unprincipled.
> >
> > This may be subjective, but I believe that it is a more honest representation of what you are doing though; weight decay is a very standard technique, and you don't need to invent a complicated justification for using it.
> >
> > To be clear, I don't have an issue with adding the weight decay term to your objective. I don't like the way you present it, but it is not a very significant issue and I am not basing my score on this issue.
> >
> > **Q3.** I am still not completely satisfied with your repsonse to this question.
> >
> > > The “uncertainty” yielded by DE totally stems from the randomness in DNNs initialization and SGD, and hence it is just some kind of stochasticity instead of the Bayesian uncertainty.
> >
> > It is not very clear what is meant by "Bayesian" uncertainty vs "just stochasticity". The blogpost argues that in approximating the Bayesian model average (predictions of BNNs on some input) you do not necessarily have to produce approximate samples from the posterior, and instead can view approximating BMA as a general integral approximation problem. In this case, deep ensembles are a sensible procedure, you use the modes of the posterior to form a good approximation of BMA from a low number of samples. Empirical experiments show that DE can indeed approximate BMA quite well in practice [1].
> >
> > So my question is still
> > > what exactly constitutes a Bayesian method, and why the proposed procedure is "more Bayesian" than the original deep ensembles?
> >
> > I encourage the authors to carefully explain these points in the text.
> >
> > **Q4.** Thank you for these experiments! It would be good to see if the uncertainty obtained by DE-GP is indeed better than the uncertainty of rDE in this case, as otherwise these results do not show a clear advantage for DE-GP.
> >
> >
> > [1] What Are Bayesian Neural Network Posteriors Really Like?
> > Pavel Izmailov, Sharad Vikram, Matthew D. Hoffman, Andrew Gordon Wilson

---

> > > ### Author Response · Authors · 2021-11-19
> > > **Thank you for the follow-up comments**
> > >
> > > Thank you for the follow-up comments. Here we address your detailed concerns.
> > >
> > > ---
> > > **Q1:** What is MC NN-GP? What exactly is the prior over functions?
> > >
> > > **A1:** As you understand, a finitely wide BNN with a Gaussian prior over the parameters induces a stochastic process, but indeed not in the form of a GP. Yet, directly using that stochastic process as prior will cause learning difficulties as we don’t have access to its densities (we may have to resort to the gradient estimators for implicit distributions). The MC NN-GP applies a Gaussian approximation to that stochastic process (by estimating the empirical covariance) and eases the learning. This approximation is expected to be accurate as the width of the BNN grows. Some previous works (Wang et al., 2019) also adopted the MC NN-GP prior.
> > >
> > > We use the MC NN-GP as the prior instead of the exact NN-GP since that we cannot efficiently, accurately estimate it when the sample size is large and the NN contains max-pooling and normalizations.
> > >
> > > ---
> > > **Q2:** Regarding the weight decay term.
> > >
> > > **A2:** Thanks for the further comments. But we still want to clarify that the L2 penalty on weights rarely appears in the previous function-space Bayesian inference works, for example, it doesn't appear in the function-space particle optimization variational inference (Wang et al., 2019). The reason is that in function-space Bayesian inference, the regularization is applied on functions (like the kernel ridge regression). And people may think that introducing an extra weight-space penalty may lead to an unprincipled posterior inference. So we resort to the posterior regularization paradigm to make the weight-space regularization tenable.
> > >
> > > We will follow your advice and update the presentation of the weight decay in the revision.
> > >
> > > ---
> > > **Q3:** What exactly constitutes a Bayesian method, and why is the proposed procedure "more Bayesian" than the original deep ensembles?
> > >
> > > **A3:** In our opinion, a Bayesian method is one that performs Bayesian inference to find the Bayesian posterior. The resultant Bayesian model average (BMA) is merely an outcome of a Bayesian method. Therefore, of course, you can think of deep ensemble as an approximation to BMA, but it doesn’t mean deep ensemble is Bayesian, *since that deep ensemble doesn’t perform Bayesian inference*. Moreover, the discovery of posterior modes in deep ensemble is totally driven by some randomness, so there is no rigorous guarantee that the resultant uncertainty is reliable. For example, for unimodal loss surfaces, the ensemble members would find the same mode and then the uncertainty disappears. Yet, standard Bayesian inference such as mean-field VI, Laplace approximation, or MCMC can still find reasonable solutions under the guide of Bayesian inference principle (e.g., minimize KL). Furthermore, Figure 1 empirically shows that the uncertainty given by deep ensemble would also degenerate when the model is significantly over-parameterized.
> > >
> > > Conclusively, the “Bayesian” mentioned by us means doing Bayesian inference and finding the Bayesian posterior, which guarantees the reliability of the outcomes. The “more Bayesian” of our method is attributed to the fact that it relies on function-space VI to find a non-degraded, reliable Bayesian posterior, instead of randomly taking some posterior modes like deep ensemble.
> > >
> > > ---
> > > **Q4:** Comparison on uncertainty estimates on CIFAR-100.
> > >
> > > **A4:** Thanks for waiting for the results! We have added the comparison on uncertainty estimates on CIFAR-100 to Appendix B.4. In particular, we have provided the error versus uncertainty plots on the combination of CIFAR-100 and SVHN test sets in Figure 13. We have depicted the results on CIFAR-100 corruptions in Figure 14. As shown, the uncertainty estimates yielded by DE-GP for OOD data are more calibrated than the baselines. DE-GP also reveals lower NLL at various levels of image skew.

---

> > > > ### Comment · Reviewer_uL2a · 2021-11-29
> > > > **Thank you for the clarifications**
> > > >
> > > > Dear authors, thank you for the further clarifications!
> > > >
> > > > Q1. Thank you for the clarifications, I would encourage you to describe this point very clearly in the text, as it is an important part of the method and is fairly confusing.
> > > >
> > > > Q2/Q3. Thank you for the clarifications on your position. As a suggestion I would recommend to (1) simplify the presentation of the weight decay term and (2) move away from the presentation idea of "making deep ensembles Bayesian" and focus more on the practical aspects of the proposed procedure, such as the practical performance (measured by accuracy or uncertainty quality) or the similarity of the predictions to the true BMA (in cases when it is possible to approximate it with high accuracy). It is not fully clear why we want to make deep ensembles more Bayesian, how to measure "how Bayesian" the result is, and why it matters.
> > > >
> > > > Q4. Thank you for adding these results, I believe they add value to the paper!
> > > >
> > > > To sum up, I think this is a borderline paper. The main strength of the paper is that the proposed method provides a useful improvement compared to standard deep ensembles and variations in terms of uncertainty estimates. On the downside, the presentation focuses primarily on "making deep ensembles Bayesian", which is a somewhat vague goal. The presentation heavily relies on the idea that "deep ensembles lack any Bayesian justification", while prior work has provided Bayesian justification for deep ensembles.

---

> > > > > ### Author Response · Authors · 2021-11-29
> > > > > **Thanks!**
> > > > >
> > > > > Dear reviewer, thanks a lot for your time and efforts! We particularly appreciate your acknowledgment of the practical value of our paper and suggestions on the presentation!
> > > > >
> > > > > We will update the paper accordingly in the next version.
> > > > >
> > > > > We believe that this paper has provided timely and valuable insights and contributions for both the deep uncertainty estimation community and the Bayesian deep learning community. We sincerely hope that the reviewer could take these contributions as a piece of evidence to increase the score.
> > > > >
> > > > > Best,
> > > > > The authors

---

> ### Author Response · Authors · 2021-11-15
> **Thank you for the valuable review (Part 1/2)**
>
> We thank the reviewer for the constructive comments and valuable feedback.  We address specific comments below and have updated the paper accordingly.
>
> ---
>
> **Q1:** The motivation of DE-GP. DE-GP is just recovering the NN-GP posterior? Why is DE-GP better?
>
> **A1:** We thank you for the comments. To clarify, we point out that
> - Our goal is to develop a Bayesian refinement of deep ensemble instead of recovering the NN-GP posterior.
> - *We have high freedom to determine which prior to use* attributed to the variational inference framework.
> - We are using the Monte Carlo NN-GPs (MC NN-GPs) associated with *finitely wide architectures* as the prior. In fact, such MC NN-GP priors correspond to the conventionally used weight-space Gaussian priors. Namely, we follow the common practice for defining priors in (Bayesian) deep learning. Similar setups can be found in some closely related works like f-POVI (Wang et al., 2019). Though the MC NN-GPs would converge to the exact NN-GPs as the model width grows (Novak et al., 2018), they are different things.
> - Even if the true posteriors are the NN-GP posteriors, *we can hardly estimate them* when facing classification problems, large data (Shi et al., 2019), and contemporary DNN architectures (Novak et al., 2018), e.g., ConvNets with max-pooling and BNs.
> - As you mentioned, the family specified by deep ensemble may actually enjoy the beneficial inductive bias of practically sized SGD-trained DNNs. Besides, DE-GP can flexibly trade off between the likelihood and the prior by tuning $\alpha$, thus possessing better performance than NN-GP in UCI regressions.
>
> We have revised the paper accordingly.
>
> ---
>
> **Q2:** Regarding Section 3.3.
>
> **A2:** We respectfully point out that our constraint is not on *"the accuracy on the train set"* but on the *true accuracy/risk* defined on the underlying distribution $\mu(x, y)$ which generates the training data. This is a general definition in the learning theory community.
>
> As a result, the first issue raised by the reviewer does not exist. We indeed guarantee the accuracy of the training data by the likelihood, but to make the trained model more *generalizable* by the constraint.
>
> As the reviewer understands, the regularization just boils down to an extra weight decay term. But, directly saying we add an extra weight-space regularization to the function-space variational inference framework would make the readers think that our approximate inference is unprincipled. Therefore, we resort to the *posterior regularization* paradigm (Ganchev et al., 2010; Zhu et al., 2014) to make the regularization tenable.
>
> We also clarify that there are extensive empirical and theoretical works showing that the generalization error of DNNs can be reduced by norm-based model capacity control (Neyshabur et al., 2015; 2017; Bartlett et al., 2017; Jiang et al., 2019).
>
>
>
> ---
> **Q3:** About “making deep ensembles Bayesian” and related work [1]:
>
> **A3:** In a typical Bayesian approach, we are interested in finding the *posterior* distribution of some random variables given the *prior belief* and the *data likelihood*. The benefits of a Bayesian approach naturally include the Bayesian uncertainty. By this definition, it is hard to interpret DE as a Bayesian approach. The “uncertainty” yielded by DE totally stems from the randomness in DNNs initialization and SGD, and hence it is just some kind of stochasticity instead of the Bayesian uncertainty.
>
> In the [post](https://cims.nyu.edu/~andrewgw/deepensembles/) and the corresponding paper (Wilson & Izmailov, 2020), the authors stated that deep ensemble is a method that approximates the *Bayesian posterior predictive distribution* -- the most important outcome of a Bayesian approach, but it is hard to judge whether the approximation is reliable or not in practice  (see Figure 1: sometimes DE yields good predictive distributions, while sometimes yields bad ones). By contrast, DE-GP leverages function-space variational inference to *explicitly* regularize the model's behaviors, which guarantees the quality of the approximate posterior predictive and the uncertainty estimates.
>
> By the way, as you mentioned, many researchers are still devoted to providing Bayesian justification for deep ensemble under certain assumptions (Lu & Van Roy, 2017; Osband et al., 2018; Pearce et al., 2020; He et al., 2020;  D’Angelo & Fortuin, 2021).
>
> We have made these clearer and discussed the related work [1] in the updated paper (See section 2 in the revision).

---

### Author Response · Authors · 2021-11-15
**Thank the reviewers and ACs for their efforts in reviewing our paper!**

We thank all the reviewers and ACs for their efforts in reviewing our paper and for providing constructive feedback. We are happy to see that Reviewer uL2a, Reviewer 5XQ4, and Reviewer zZzc appreciate the extent of our experiments and the demonstrated effectiveness of our approach. We are encouraged by the acknowledgment of the significance of our contribution to the community by Reviewer 5XQ4 and Reviewer zZzc. We have addressed the detailed comments to strengthen our paper. In summary, here are the main changes that we made to the paper:

- Clarified that it is free to choose various priors and DE-GP is not just recovering the NN-GP posterior. (Reviewer uL2a)
- Strengthened the discussion on the pain points of deep ensemble. (Reviewer uL2a, Reviewer Bycy)
- Added CIFAR-100 results in Appendix B.4. (Reviewer uL2a)
- Fixed some presentation issues. (Reviewer 5XQ4)
- Added an ablation study on $\alpha$ in Appendix B.5. (Reviewer 5XQ4)
- Fixed typos and grammatical errors; improved the introduction section and the related works section; and revised the references. (Reviewer zZzc)
- Made our contributions clearer. (Reviewer Bycy, Reviewer hBTZ)
- Clarified the limitations of NN-GP, RMS, and NTKGP. (Reviewer Bycy)
- Revised the title as “Deep Ensemble as a Gaussian Process Approximate Posterior”. (Reviewer hBTZ)
- Clarified the posterior regularization in function space. (Reviewer hBTZ)
- Performed more trials for Table 1 and Figure 7. (Reviewer hBTZ)
- Clarified the baselines in the contextual bandit experiments. (Reviewer hBTZ)

Overall, we have carefully addressed the main concerns in detail. We hope you might find the response satisfactory. We will be very happy to clarify any further concerns (if any).

---

### Author Response · Authors · 2021-11-21
**Look forward to further feedback**

Dear reviewers,

We thank you again for the valuable comments. We are looking forward to hearing from you about any further feedback.

If you find the response satisfactory, we hope you might view this as a sufficient reason to further raise your score.

If you still have questions about our paper, we are willing to discuss them with you and improve our paper.

Best,

Authors

---

### Author Response · Authors · 2021-12-01
**Regarding the argument "making deep ensembles Bayesian"**

We thank Reviewer uL2a and Reviewer Bycy again for the valuable comments. But we are sorry to still hear that the argument "make deep ensembles Bayesian" is not well-motivated.

We clarify that this argument is widely accepted in the Bayesian deep learning community. In particular, Pearce et al., 2020 stated that *"Ensembling NNs provides an easily implementable, scalable method for uncertainty quantification, however, it has been criticised for **not being Bayesian**"*; He et al., 2020 stated that *"Despite the recent interest in BNNs, it has been shown empirically that deep ensembles, which **lack a principled Bayesian justification**, outperform existing BNNs in terms of uncertainty quantification and out-of-distribution robustness"*. They are in line with this paper and motivate us to develop a more practical Bayesian variant of deep ensemble without relying on limited assumptions as they have made.

Though interpreting ensembles as an approximation of Bayesian posterior predictive is correct, a concurrent submission to ICLR 2022 (Anonymous, 2022) has made a similar argument to ours: *"obtaining individual ensemble members via maximum a posteriori probability (MAP) estimation ...  **has arbitrary bad approximation guarantees**. For example, the resulting
approximation can be poor in the case when the true posterior distribution is unimodal."*

All of these are aimed to evidence that deep ensemble is far from a Bayesian inference approach and there will be serious drawbacks.

Anyway, we will revise the argument "making deep ensembles Bayesian" in the next version. Basically, **we will first clarify that deep ensemble relies on the multi-modal loss landscape and random initialization which results in uncontrolled posterior approximation (i.e., there may be arbitrary bad approximation guarantees). Thus, we suggest relating deep ensemble with Bayesian inference to achieve better posterior approximation and to enjoy reliable Bayesian uncertainty. We propose one modification to standard deep ensemble -- defining a GP with it -- and perform function-space variational inference to realize the above goal. We prove that our DE variant makes more conservative predictions and offers more faithful uncertainty than the standard deep ensemble.**

We hope you might find our reply satisfactory. We will improve the paper in the final version.

Best,

The authors

> Uncertainty in Neural Networks: Approximately Bayesian Ensembling. Pearce et al., AISTATS 2020.

> Bayesian Deep Ensembles via the Neural Tangent Kernel. He et al., NeurIPS 2020.

> Greedy Bayesian Posterior Approximation with Deep Ensembles. Anonymous, ICLR 2022 under review.

---

### Decision · Program_Chairs · 2022-01-20

**Decision:**

Reject

**Comment:**

This paper proposes to use deep ensembles to parameterize a variational Gaussian process posterior, and uses an additional L2 penalty on parameters of the neural networks, and an (MC) NN-GP prior (although the prior is a free design choice). Reviewers appreciated aspects of the paper, finding there to be a minor improvement in uncertainty calibration over regularized deep ensembles, and nice results for the contextual bandit experiments. Ultimately, however, after a healthy and active exchange between reviewers and authors, four out of five reviewers are voting to reject the paper. There is a belief that the paper can be substantially improved from its current form, by carefully accommodating reviewer feedback, but it is not currently at a stage ready for publication.

There were common themes in the concerns expressed by several reviewers. Many reviewers found the technical contributions incremental. Parametrizing a GP using deep ensembles, or adding L2 regularization, is not itself a major technical contribution, and the variational framework leans heavily on Sun et. al (2019) and work that came before it from Titsias (2009). Similarly, the theoretical contributions were found to be incremental.

These concerns about the technical contributions may have been counterbalanced if the experimental results had been outstanding or the framing of the paper perceived to be very clear and well justified. However, the experimental results had a mixed reception, with several reviewers noting accuracy was not in fact much better than the simpler regularized deep ensembles, despite some improvements in uncertainty calibration. One reviewer liked the bandit experiments, but wished there was a deeper exploration of this application domain. The current experimental results do not seem to warrant the relative complexity of the approach over simple regularized deep ensembles.

Additionally, several reviewers found the framing and presentation of the paper needing significant work. The introduction of the L2 regularization terms, for example, was perceived to be overly complex, involving several steps that were not well-motivated.

Several reviewers also found the motivation about making deep ensembles Bayesian unconvincing. A procedure being sensitive to initialization, or unreliable in certain settings, does not mean it does not perform approximate Bayesian inference. For example, variational methods and Laplace approximations can depend on initialization, and could get stuck in poor local optima. Quoting papers referring to deep ensembles as non-Bayesian is also not an argument in itself. The blog post linked by a reviewer is clearly pushing back against these claims, and does address points raised in the discussion, such as unimodal approximations and theoretical guarantees. As reviewers have also noted, several papers have now provided plain deep ensembles with a Bayesian justification, and these papers should be acknowledged. It could be reasonable to argue that your paper makes deep ensembles _more_ Bayesian, and you could potentially try to measure this claim in a concrete way. Or you could simply argue that your approach helps reduce sensitivity to initialization, and represents solutions with lower posterior density, which can be helpful practical contributions and don't need to be tied to claims about the method being Bayesian.

Please thoughtfully reflect on the reviewer comments in updated versions of the paper. The reviewers put a lot of effort into providing feedback and engaging during the rebuttal period. While the paper has some nice features, there is significant room for improvement on several fronts: technical innovation, experimental investigation, and framing. Improving the framing will help, but working further to also address other concerns will likely be needed to sway reviewers.